# FedUSD: Unbiased Synthetic Data for Federated Learning

**Weiying Xie** [1]  **Chenhe Hao** [1]  **Haozhi Shi** [1]  **Jitao Ma** [1]  **Daixun Li** [1]  **Jiazhe Li** [1]  **Hengyi Wang** [1]  **Leyuan Fang** [2]  **Yunsong Li** [1]

## Abstract

Aggregation-Free Federated Learning enables joint training by sharing synthetic data, aiming to eliminate data heterogeneity across clients. However, existing methods fail to explicitly separate the principal and residual components of dataset, leading to biased synthetic data. In this paper, we propose a novel Unbiased Synthetic Data optimization method FedUSD for Aggregation-Free Federated Learning, which is achieved by exploring the High-energy Orthogonal Base (HOB) and variance of dataset in feature space. Our FedUSD is inspired by the discovery that principal component concentrates in HOB while residual component independently reflects in variance, regardless of networks. Based on the observation, we develop a method that mathematically optimizes synthetic data by matching both HOB and variance with those of real data. Besides, we experimentally show the superior effectiveness of leveraging HOB and variance to separately extract the principal and residual components over existing methods. We also theoretically prove that FedUSD achieves unbiased synthetic data and thus convergence. Without introducing any constraints, FedUSD thereby yields significant improvements over the state-of-the-arts in terms of global model performance, under equivalent communicational costs. For example, on the SVHN dataset, FedUSD improves 6.74% to 30.82% which is higher than others with Dirichlet coefficient $\alpha = 0.01$.

## 1. Introduction

Federated Learning (FL), as a distributed machine learning paradigm, has been widely explored in fields such as

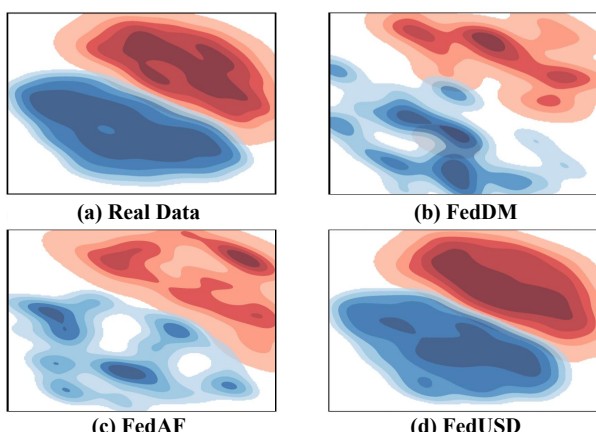

*Figure 1.* Comparison of *t*-SNE impact on CIFAR-10 under different Aggregation-Free Federated Learning frameworks. The synthetic data generated by our proposed FedUSD better approximates the distribution of real data.

computer vision (Chen et al., 2023b; Wu et al., 2022), medical analysis (Koutsoubis et al., 2024), and autonomous driving (Fanì et al., 2023). However, data heterogeneity is an inherent challenge in FL (Wang et al., 2024a). Specifically, data collected by different clients often follow a non-independently and identically distribution (Non-IID) due to divergent preferences, which ultimately leads to performance degradation.

Currently, numerous works aim to address data heterogeneity by designing aggregation scheme (Hsu et al., 2019; Lin et al., 2020; Wang et al., 2020; Li et al., 2021). Typical works either build novel strategies to enhance the aggregation phase (Wang et al., 2020; Zhang et al., 2024b), or adaptively adjust aggregation using learned local information as a prior (Wang et al., 2024a; Chen et al., 2023a; Wang & Ji, 2023; Zhang et al., 2022). However, aggregation schemes inherently struggle to eliminate data heterogeneity as they fail to balance data distribution, thereby limiting the global model performance. In contrast, recently Aggregation-Free FL (AFFL) methods significantly eliminate data heterogeneity (Xiong et al., 2023; Wang et al., 2024c). On the basis of dataset distillation (Zhang et al., 2025; Du et al., 2024; Zhao & Bilen, 2021), each client generates class-balanced synthetic data for majority classes, and each synthetic-set

---

[1]Xidian University, Xi'an, China [2]Hunan University, Changsha, China. Correspondence to: Chenhe Hao <hch@stu.xidian.edu.cn>.

*Proceedings of the 43$^{rd}$ International Conference on Machine Learning*, Seoul, South Korea. PMLR 306, 2026. Copyright 2026 by the author(s).

achieves similar performance with the real data. AFFL eliminates data heterogeneity by training on such a synthetic-set at server. While certain AFFL methods have shown encouraging performance, the synthetic data still struggles to capture the underlying characteristics of the real data. Currently, existing methods typically optimize synthetic data based on distribution matching, with the purpose of approximating the real data. The core idea of distribution matching is to guide optimization by balancing the contributions of principal and residual component. When representing class-level principal component, most existing approaches simply use the mean of hidden representations. However, mean representation contains both class-level principal component and part of the uncontrollable residual component, which introduces a bias during synthetic data optimization (see Figure 1). As a result, the bias will lead to information loss and ultimately harm the performance of global model.

In this paper, we propose an **Unbiased Synthetic Data** optimization method **FedUSD** for **Federated Learning** to mitigate data heterogeneity across clients. We innovatively employ High-energy Orthogonal Base (HOB) and variance as essential elements of FedUSD, since they accurately capture real data characteristics in feature space. In particular, HOB extracts class-level principal component while variance independently captures residual component. Besides, we use Split-and-Expand Augmentation (SEA) (Zhao et al., 2023) to increase the information capacity of the synthetic data. These designs make FedUSD a potential method applicable to various FL scenarios. Specifically, we first initialize the synthetic data on each client by using SEA. Next, each client optimizes synthetic data by matching the HOB and variance with those of real data. Then, each client sends its synthetic data to server. Through this mechanism, data privacy protection is substantially strengthened (**shown in Appendix**). Finally, server receives the synthetic data, restores each sub-synthetic data to the real data size, and uses them for continue training. Extensive experiments are consistent with the theoretical guarantee of unbiased optimization, which demonstrates that FedUSD exhibits better convergence as well as superior generalization. It enables FedUSD to outperform other methods across different heterogeneity settings on various datasets and architectures.

Our main contribution are summarized as:

- We propose FedUSD, a novel Aggregation-Free FL method to eliminate data heterogeneity. To our best knowledge, our FedUSD is the first work that designs an unbiased synthetic data optimization in FL.

- We pioneeringly optimize the synthetic data with HOB and variance which can independently extract the class-level principal and residual component of dataset in feature space, thereby greatly enhancing the ability of

synthetic data to represent real data.

- We propose the first unified convergence theory for AFFL and FL under the same conditions, in which AFFL consistently achieves better convergence. We also derive the crossover round to explain its gains, and prove that FedUSD satisfies differential privacy.

- Extensive experiments have demonstrated our high communicational efficiency and better performance over existing methods, while rigorous theoretical analysis also shows that FedUSD achieves unbiased optimization. Besides, our designed ablation experiments also show effectiveness in each of our module.

## 2. Related Works

### 2.1. Federated Learning

FL still remains many open problems to solve (Ye et al., 2025; Li et al., 2020a; McMahan et al., 2017). In this work, we focus on data heterogeneity in FL. Previous researches have addressed it by designing aggregation schemes (Zhu et al., 2021a; Wang et al., 2020; 2024b; Zhu et al., 2021b; Pillutla et al., 2022). FedProx introduces a proximal regulation term to constrain the deviation between the local and global models (Li et al., 2020b). SCAFFOLD mitigates local updated biases by introducing control variates (Karimireddy et al., 2020). While these aggregation schemes partially mitigate the impact of data heterogeneity on global model optimization, they fail to balance the data distribution, thus limiting global model performance.

Recently, some works introduce the AFFL to deal with data heterogeneity (Shi et al., 2025). On the basis of dataset distillation, they generate distribution-balanced synthetic data on clients and send it to the server for training (Liu et al., 2023). FedDM optimizes synthetic data by matching the mean and variance (Xiong et al., 2023). Building upon it, FedAF improves the performance of global model by incorporating a regularization term (Wang et al., 2024c). However, existing methods ignore flaws in their distillation strategies, which our FedUSD focus to solve.

### 2.2. Dataset Distillation

Dataset distillation is a method of dataset compression. It optimizes a small synthetic-set to match the performance of the real data, thereby reducing computational overhead (Sachdeva & McAuley, 2023; Wang et al., 2022; Du et al., 2024). Some works aim to guide the optimization of synthetic data by using metrics obtained during training (Zhao & Bilen, 2021; Cazenavette et al., 2022). For example, DC generates the synthetic data by matching the gradients between real data and synthetic data (Zhao et al., 2021). However, these methods rely on results during training, thus

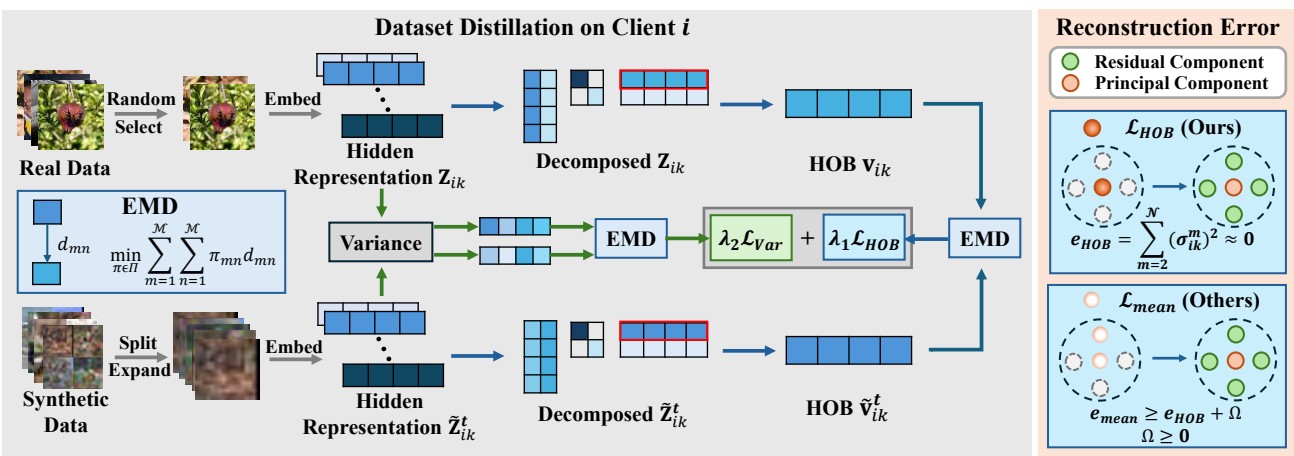

*Figure 2.* Overview of FedUSD. The right part illustrates the key of FedUSD: Compared to mean, using HOB to represent the principal component results in smaller reconstruction error verging to zero. Thus, by collaboratively matching HOB and variance, we optimize the synthetic data to better approximate the distribution of the real dataset.

typically incurring high computational overhead, which is infeasible in FL. Some dataset distillation methods optimize synthetic data by matching its distribution with that of the real data (Zhao & Bilen, 2023; Zhang et al., 2024a). By requiring no training but only feature embedding, they are highly efficient and suitable for FL, that is also the basis of AFFL (Zhao et al., 2023). Most methods optimize synthetic data by adjusting the mean-variance ratio, however, the mean includes uncontrollable class-level residual components, which results in a biased optimization. In this paper, we design new unbiased metrics to represent class-level principal and residual components, thereby guiding unbiased synthetic data optimization for FL.

## 3. Methodology

### 3.1. Preliminaries

**Federated Learning**. In the sequel, we denote vectors, matrices and tuples by bold-faced letters. We use $\|\cdot\|_p$ as the $\ell_p$ norm of vectors or matrices, and $[n] = 1, 2, ..., n$. In our FL set up, the real dataset $D = \bigcup_{i=1}^{N} D_i$ is divided among $N$ clients in a disjoint and Non-IID way, and each client $i$ has a data chunk $D_i$ consisted of $n_i = \sum_{k \in [\mathcal{K}]} n_{ik}$ samples $\{(\mathbf{x}_{ij}, y_{ij})\}_{j=1}^{n_i}$, here $\mathbf{x}_{ij}$ denotes the $j^{th}$ data point on client $i$, $y_{ij} \in [\mathcal{K}]$ represents its corresponding target, and $n_{ik}$ represents the number of samples belonging to target $k$ on client $i$. Typically, the global model $\mathbf{w}$ is updated as:

$$\arg \min_{\mathbf{w}} F(\mathbf{w}) = \arg \min_{\mathbf{w}} \sum_{i=1}^{N} p_i F_i(\mathbf{w}, D_i), \quad (1)$$

$$F_i(\mathbf{w}, D_i) = \frac{1}{|D_i|} \sum_{i=1}^{|D_i|} \ell_i(\mathbf{w}, \mathbf{x}_{ij}, y_{ij}), \quad (2)$$

where $F_i(\mathbf{w}, D_i)$ denotes the loss of client $i$, and $\ell_i(\cdot)$ is the training loss. Weight coefficient $p_i$ associated with client $i$ is proportional to $|D_i|$ and normalized by $p_i = \frac{|D_i|}{|D|}$.

**Dataset Distillation with Distribution Matching**. Suppose a client $i$ is assigned to optimize a synthetic dataset $\widetilde{D}_i^t$ at communication round $t$, and $|\widetilde{D}_i^t| << |D_i|$. The client initializes the synthetic data $\widetilde{\mathbf{x}}_{ij}$ by sampling from real dataset or Gaussian noise. Then, clients employ a feature extractor $h_{\mathbf{w}}(\cdot)$ to extract the hidden representation of data. The means of hidden representations are computed as:

$$\mu_{ik} = \frac{1}{n_{ik}} \sum_{j=1}^{n_{ik}} h_{\mathbf{w}}(\mathbf{x}_{ij,k}), \; \widetilde{\mu}_{ik}^t = \frac{1}{\widetilde{n}_{ik}} \sum_{j=1}^{\widetilde{n}_{ik}} h_{\mathbf{w}}(\widetilde{\mathbf{x}}_{ij,k}). \; (3)$$

Here, $\mathbf{x}_{ij,k}$ and $\widetilde{\mathbf{x}}_{ij,k}$ represent the $j$-th sample of class $k$ from $D_i$ and $\widetilde{D}_i^t$ respectively. $\widetilde{n}_{ik}$ denotes the number of samples under class $k$ in $\widetilde{D}_i^t$. After that, client $i$ can optimize the $\widetilde{D}_i^t$ by Distribution Matching (DM) loss as:

$$\arg \min_{\widetilde{D}_i^t} \mathcal{L}_{DM}(\widetilde{D}_i^t, D_i) = \arg \min_{\widetilde{D}_i^t} \sum_{k \in \mathcal{K}} \|\mu_{ik} - \widetilde{\mu}_{ik}^t\|_2. \; (4)$$

### 3.2. Process Description

The overall framework of FedUSD is illustrated in Figure 2. For each communication round $t$, clients first engage in collaborative dataset distillation. Then they share the synthetic data with the server. Finally, the server expands the synthetic data by using Split-and-Expand Augmentation (SEA) (Zhao et al., 2023) and updates the global model.

**Principal and Residual Component Matching**. In DM, using only mean would bias synthetic data toward class centers, reducing diversity and generalization. Hence, to

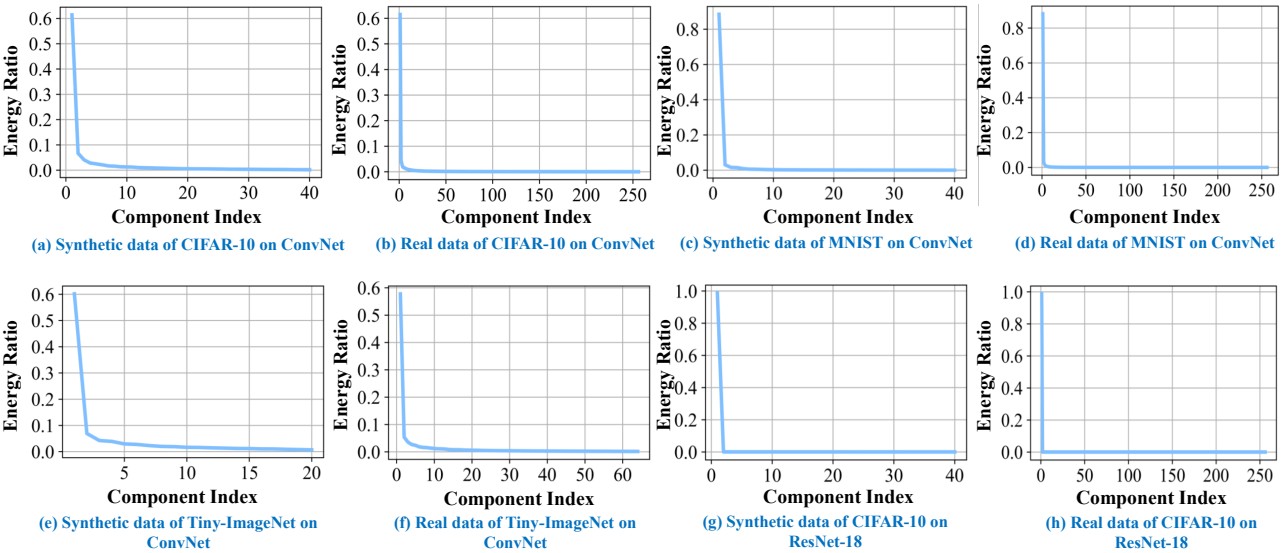

*Figure 3.* Energy spectrum of synthetic data and real data across different networks and datasets. In all cases, the majority of energy consistently concentrates in the first orthogonal basis.

accurately fit the distribution of real data, we optimize the synthetic data by matching both principal and residual components. Following it, we minimize the gaps in principal and residual components between synthetic and real data, which re-defines unbiased synthetic data optimization:

$$\widetilde{D}_{ik}^{t*} = \arg\min_{\widetilde{D}_{ik}^{t}} \lambda_1 \|\mathbf{z}_{ik}^{\mathrm{PC}} - \widetilde{\mathbf{z}}_{ik,t}^{\mathrm{PC}}\| + \lambda_2 \|\mathbf{z}_{ik}^{\mathrm{RC}} - \widetilde{\mathbf{z}}_{ik,t}^{\mathrm{RC}}\|, \quad (5)$$

where $\mathbf{z}_{ik}^{\mathrm{PC}}$ and $\widetilde{\mathbf{z}}_{ik,t}^{\mathrm{PC}}$ represent the principal component of $D_{ik}$ and $\widetilde{D}_{ik}^{t}$, while $\mathbf{z}_{ik}^{\mathrm{RC}}$ and $\widetilde{\mathbf{z}}_{ik,t}^{\mathrm{RC}}$ representing the residual component of $D_{ik}$ and $\widetilde{D}_{ik}^{t}$ respectively. $\widetilde{D}_{ik}^{t*}$ represents the optimal synthetic dataset on client $i$ for class $k$, $k$ denotes the classes with a selected sample size greater than the synthetic Images Per Class (IPC) setting.

**HOB Representation**. Existing methods usually represent the principal component of a class distribution by its mean feature. However, the mean feature inevitably mixes the dominant direction with residual, which may introduce bias into synthetic data optimization. To obtain a more explicit representation of the principal structure, we decompose the hidden representations of real and synthetic data as:

$$\mathbf{Z}_{ik} = \mathbf{U}_{ik}\boldsymbol{\Sigma}_{ik}\mathbf{V}_{ik}^{\top}, \ \widetilde{\mathbf{Z}}_{ik}^{t} = \widetilde{\mathbf{U}}_{ik}^{t}\widetilde{\boldsymbol{\Sigma}}_{ik}^{t}\widetilde{\mathbf{V}}_{ik}^{t\top}, \qquad (6)$$

where $\mathbf{Z}_{ik}$ and $\widetilde{\mathbf{Z}}_{ik}^{t}$ denote the hidden representation sets of $D_{ik}$ and $\widetilde{D}_{ik}^{t}$, respectively. The row vectors of $\mathbf{V}_{ik}^{\top}$ and $\widetilde{\mathbf{V}}_{ik}^{t\top}$ form orthogonal bases, while $\boldsymbol{\Sigma}_{ik}$ and $\widetilde{\boldsymbol{\Sigma}}_{ik}^{t}$ characterize the energy assigned to each orthogonal basis direction. We define the first orthogonal basis with the largest singular value as the high-energy orthogonal base (HOB), which is used to describe the principal direction of the class-conditional feature distribution.

It is worth noting that singular vectors have inherent sign ambiguity, i.e., $\mathbf{v}$ and $-\mathbf{v}$ represent the same direction. Therefore, before computing the HOB matching loss, we adopt a sign-invariant treatment to align the synthetic HOB with the real HOB. Specifically, the matching is performed after selecting the orientation that gives the smaller discrepancy between the two directions. In this way, the loss measures the difference between principal directions rather than being affected by arbitrary sign flips in SVD.

In addition to HOB matching, we also match the feature-wise variance to preserve the residual dispersion of the class distribution. For both HOB and variance matching, we use the Earth Mover's Distance (EMD) to treat these vectors as feature-space distributions and compare their overall shapes. Compared with purely coordinate-wise distances, EMD captures global mass relocation across feature dimensions, which is better suited to deep representations where different feature dimensions may be coupled or redundant. Here, the dimension parameter in Eq. (14) denotes the dimensionality of the HOB or variance vector being matched.

As shown in Figure 3, the energy is concentrated in the first orthogonal basis ($\mathbf{v}_{ik}$ and $\widetilde{\mathbf{v}}_{ik}^{t}$). Thus we regard the High-energy Orthogonal Base (HOB) $\mathbf{v}_{ik}$ and $\widetilde{\mathbf{v}}_{ik}^{t}$ as unbiased extractor of the principal component for data, and propose it as a more expressive substitute for the mean.

To prove it, we formulate the principal component representation as a rank-1 reconstruction problem. We firstly introduce an assumption utilized for our analysis.

**Assumption 3.1** (**Energy Concentration**). For hidden representation set derived from dataset, the energy is concentrated in the first orthogonal basis (HOB) of input space,

and satisfies: $\sigma_{ik}^1 >> \sigma_{ik}^m, m \neq 1, \sigma_{ik}^m = \mathbf{\Sigma}_{ik}[m][m]$.

Due to consistent data distributions, hidden representations of same-class samples tend to be approximate low-rank, consisting of a dominant prototype direction with small residuals. Consequently, the spectral energy is dominated by the largest singular value. It is also observed in Figure 3, supporting the validity of Assumption 3.1.

For any $\mathbf{Z}_{ik}$ belonging to the same class $k$ on client $i$, rank-1 reconstruction problem can be described as:

$$\min_{g_{ik}, \beta_{ik}^1, ..., \beta_{ik}^{n_{ik}}} \|\mathbf{Z}_{ik} - \beta_{ik} g_{ik}^\top\|_F^2, \beta_{ik} = [\beta_{ik}^1, ..., \beta_{ik}^{n_{ik}}]^\top, \quad (7)$$

where, $g_{ik}$ is the representation of principal component, $\beta_{ik}$ is the projection of each hidden representation onto $g_{ik}$.

The reconstruction error when using the HOB is:

$$e_{HOB} = \sum_{m=2}^{\mathcal{N}} (\sigma_{ik}^m)^2, \mathcal{N} = \text{rank}(\mathbf{Z}_{ik}). \quad (8)$$

The reconstruction error when using the mean is:

$$e_{mean} \geq \sum_{m=2}^{\mathcal{N}} (\sigma_{ik}^m)^2 + ((\sigma_{ik}^1)^2 - \|\mathbf{Z}_{ik}\mu_{ik}^\top\|_2^2). \quad (9)$$

Because of $\|\mathbf{Z}_{ik}\mu_{ik}^\top\|_2^2 \leq (\sigma_{ik}^1)^2$, we can get the theorem (**proof is presented in Appendix**).

**Theorem 3.1** (**Less Reconstruction Error for HOB**). *The HOB yields a smaller reconstruction error than the mean:*

$$e_{HOB} \leq e_{HOB} + \Omega \leq e_{mean}, \ \Omega \geq 0. \quad (10)$$

*The equality is achieved only if $\mathbf{v}_{ik}\|\mu_{ik}$ and hidden representations have approximately the same magnitude. Under the Assumption 3.1, the $e_{HOB}$ is close to 0.*

**Optimization Function.** Based on Theorem 3.1, we use the HOB to extract the principal feature direction with reduced mean induced bias, and use feature wise variance to characterize the residual dispersion of class conditional representations. By jointly matching these two statistics, the synthetic data can better preserve both the principal and residual structures of the real data. Moreover, instead of using the Euclidean distance that enforces strict coordinate-wise matching, we adopt the Wasserstein-1 distance, i.e., EMD, to align the overall distributional shape of the HOB and variance vectors in the feature space. This is because deep representations usually contain coupled and redundant feature dimensions, where exact coordinate-wise correspondence may be less important than the global way in which feature energy is distributed across dimensions. Then, the proposed USD Loss $\mathcal{L}_{USD}$ is defined as:

$$\mathcal{L}_{USD}(\widetilde{D}_{ik}^t, D_{ik}) = \lambda_1 \mathcal{L}_{HOB} + \lambda_2 \mathcal{L}_{Var}, \quad (11)$$

---

**Algorithm 1** Unbiased Synthetic Data for FL (FedUSD)

**Input**: Dataset $D_i$; Network $\mathbf{w}$; SEA factor $l$.
**Output**: Global model $\mathbf{w}$.
**Parameters**: Communication round $T$; Local optimization round $E$; Global training round $R$.

1: **for** $t = 0, 1, ..., T - 1$ **do**
2:    **Server executes:**
3:      Send global model weights $\mathbf{w}^{(t)}$ to all clients.
4:    **for** each client $i$ in parallel **do**
5:      Initial the synthetic data $\widetilde{D}_i^t$ by using SEA.
6:      **for** $e = 0, 1, ..., E - 1$ **do**
7:        Split and Expand the $\{\widetilde{\mathbf{x}}_{ij}\}$ in $\widetilde{D}_i^t$ to $\{\widetilde{\mathbf{x}}_{ij}^{l \times l}\}$.
8:        Calculate $\mathcal{L}_{HOB}$ by using **Equation 12**.
9:        Calculate $\mathcal{L}_{Var}$ by using **Equation 13**.
10:       Update $\widetilde{D}_i^t$ by minimizing **Equation 11**.
11:      **end for**
12:      Send the $\widetilde{D}_i^t$ to server.
13:    **end for**
14:    **for** $r = 0, 1, ..., R - 1$ **do**
15:      **Server executes:**
16:        Construct the $\widetilde{D}^t$ by using **Equation 16**.
17:        Split and Expand the $\{\widetilde{\mathbf{x}}_{ij}\}$ in $\widetilde{D}^t$ to $\{\widetilde{\mathbf{x}}_{ij}^{l \times l}\}$.
18:        Update the global model $\mathbf{w}^{(t)}$ by using $\widetilde{D}^t$.
19:    **end for**
20: **end for**

---

$$\mathcal{L}_{HOB} = \text{EMD}(\mathbf{v}_{ik}, \widetilde{\mathbf{v}}_{ik}^t), \quad (12)$$

$$\mathcal{L}_{Var} = \text{EMD}(\psi_{ik}, \widetilde{\psi}_{ik}^t), \quad (13)$$

where $\psi_{ik} = \text{Var}(\mathbf{Z}_{ik})$ denotes the variance of the hidden representation set $\mathbf{Z}_{ik}$, and EMD can be calculated as:

$$\text{EMD}(\mathbf{v}_{ik}, \widetilde{\mathbf{v}}_{ik}^t) = \min_{\pi \in \Pi} \sum_{m=1}^{\mathcal{M}} \sum_{n=1}^{\mathcal{M}} \pi_{mn} d_{mn}, \quad (14)$$

where $\mathbf{v}_{im,k} \in \mathbf{v}_{ik}, \widetilde{\mathbf{v}}_{in,k}^t \in \widetilde{\mathbf{v}}_{ik}^t, \pi_{mn} \geq 0$ is the mass moved from $\mathbf{v}_{im,k}$ to $\widetilde{\mathbf{v}}_{in,k}^t, d_{mn} = \|\mathbf{v}_{im,k} - \widetilde{\mathbf{v}}_{in,k}^t\|$ is unit transport cost, and $\Pi = \{\pi \in R_{\geq 0}^{\mathcal{M} \times \mathcal{M}} | \pi \mathbf{1} = \frac{1}{\mathcal{M}} \mathbf{1}, \pi^\top \mathbf{1} = \frac{1}{\mathcal{M}} \mathbf{1}\}$ forces uniform row/column sums to $\frac{1}{\mathcal{M}}$ (a scaled doubly stochastic plan that conserves mass on both sets).

By using $\mathcal{L}_{USD}$, we get the unbiased synthetic dataset:

$$\widetilde{D}_{ik}^{t*} = \arg\min_{\widetilde{D}_{ik}^t} \mathcal{L}_{USD}(\widetilde{D}_{ik}^t, D_{ik}). \quad (15)$$

**Global Model Optimization**. Instead of optimizing the global model via Equation 1, each client uploads their local synthetic data $\widetilde{D}_i^t$ to the server after optimizing them. Synthetic data satisfy the differential privacy requirements and thus do not leak privacy (**proof is presented in Appendix**). To reduce computation, the server trains only on synthetic

data from the current and previous $\mathbb{T} - 1$ rounds:

$$\widetilde{D}^t = \bigcup_{m=0}^{\mathbb{T}-1} \bigcup_{i=1}^{N} \widetilde{D}_i^{t-m}. \tag{16}$$

Then global model $\mathbf{w}^{(t)}$ is updated by $\widetilde{D}^t$:

$$\mathbf{w}^{(t)*} = \arg\min_{\mathbf{w}^{(t)}} F(\mathbf{w}^{(t)}, \widetilde{D}^t). \tag{17}$$

**Split-and-Expand Augmentation**. To enhance the synthetic features, we use SEA (Zhao et al., 2023). Since fine-grained details are typically lost during feature extraction, we discard the fine details before feature extraction while preserving semantics. It increases extractable features without enlarging the data. Specifically, for each $\widetilde{\mathbf{x}}_{ij}$, we split it into $l \times l$ equal pieces and expand each sub-synthetic data to size of $\mathbf{x}_{ij}$ via differentiable augmentation:

$$\widetilde{\mathbf{x}}_{ij}^1, \widetilde{\mathbf{x}}_{ij}^2, ..., \widetilde{\mathbf{x}}_{ij}^{l \times l} = \text{Expand}(\text{Split}(\widetilde{\mathbf{x}}_{ij}, l)). \tag{18}$$

# 4. Convergence Analysis

In this section, we analyze the convergence of AFFL and FL. It reveals that AFFL can give faster converge and reduce the gap between $\mathbf{w}^* = \arg\min_{\mathbf{w}} F(\mathbf{w})$ and $\lim_{t \to \infty} \mathbf{w}^{(t)}$.

## 4.1. Assumptions and Definitions

We begin by introducing four assumptions (Stich, 2019; Basu et al., 2019; Li et al., 2020c; Cho et al., 2022) and two definitions (Li et al., 2020c; Cho et al., 2022) to support the analysis that follows.

**Assumption 4.1 (L-smooth).** $F_1, ..., F_N$ are all L-smooth, i.e., for all $\mathbf{v}$ and $\mathbf{w}$, we have $F_i(\mathbf{v}) \leq F_i(\mathbf{w}) + (\mathbf{v} - \mathbf{w})^T \nabla F_i(\mathbf{w}) + \frac{L}{2} \|\mathbf{v} - \mathbf{w}\|_2^2$.

**Assumption 4.2 ($\mu$-strongly Convex).** $F_1, ..., F_N$ are all $\mu$-strongly convex, i.e., for all $\mathbf{v}$ and $\mathbf{w}$, we have $F_i(\mathbf{v}) \geq F_i(\mathbf{w}) + (\mathbf{v} - \mathbf{w})^T \nabla F_i(\mathbf{w}) + \frac{\mu}{2} \|\mathbf{v} - \mathbf{w}\|_2^2$.

**Assumption 4.3 (Bounded Variance of Gradients for FL).** For the mini-batch $\xi_i$ uniformly sampled at random from $D_i$, the resulting gradient is unbiased, that is, $\mathbb{E}[\mathbf{g}_i(\mathbf{w}_i, \xi_i)] = \nabla F_i(\mathbf{w}_i)$. Also, the variance of stochastic gradients are bounded: $\mathbb{E}\|\mathbf{g}_i(\mathbf{w}_i, \xi_i) - \nabla F_i(\mathbf{w}_i)\|^2 \leq \sigma^2$ for $i = 1, ..., N$. $\mathbf{g}_i(\mathbf{w}_i, \xi_i) = \nabla F_i(\mathbf{w}_i, \xi_i)$.

**Assumption 4.4 (Bounded Stochastic Gradient).** The expected squared norm of stochastic gradients is uniformly bounded, i.e., $\mathbb{E}\|\mathbf{g}_i(\mathbf{w}_i, \xi_i)\|^2 \leq G^2$ for $i = 1, ..., N$.

**Definition 4.1 (Degree of Non-iid).** For the global optimum of $\mathbf{w}^* = \arg\min_{\mathbf{w}} F(\mathbf{w})$ and local optimum $\mathbf{w}_i^* = \arg\min_{\mathbf{w}} F_i(\mathbf{w})$, we define the degree of non-iid as:

$$\Gamma \triangleq F^* - \sum_{i=1}^{N} p_i F_i^* = \sum_{i=1}^{N} p_i(F_i(\mathbf{w}^*) - F_i(\mathbf{w}_i^*)). \tag{19}$$

This definition was first introduced by Li *et al*. (Li et al., 2020c). $\Gamma$ reveals the degree of heterogeneity in the data distribution. Usually, larger $\Gamma$ implies higher heterogeneity.

**Definition 4.2 (Client Selection Bias).** For any $i \in \mathcal{S}$,

$$\rho(\mathcal{S}, \mathbf{w}') = \frac{\mathbb{E}_{\mathcal{S}}[\frac{1}{m} \sum_{i \in \mathcal{S}}(F_i(\mathbf{w}') - F_i^*)]}{\sum_{i=1}^{N} p_i(F_i(\mathbf{w}') - F_i(\mathbf{w}_i^*))}, \tag{20}$$

$\mathcal{S}$ represents a set of clients who participate in aggregation at each epoch. Then, we define two related metrics that are independent of $\mathbf{w}'$:

$$\overline{\rho} \triangleq \min_{\mathbf{w}'} \rho(\mathcal{S}, \mathbf{w}'), \tag{21}$$

$$\widetilde{\rho} \triangleq \rho(\mathcal{S}, \mathbf{w}^*). \tag{22}$$

The definition was introduced by Cho *et al*. (Cho et al., 2022). All $\rho$ are only related to client selection strategy, independent of dataset distillation, and $\overline{\rho} \leq \widetilde{\rho}, \overline{\rho} \leq 1$.

## 4.2. Main Convergence Result

We present the convergence results in terms of $\Gamma$ and client selection bias $\overline{\rho}, \widetilde{\rho}$ (**proof is presented in Appendix**).

**Theorem 4.1 (Convergence of FL).** *Under Assumptions 4.1 to 4.4, for learning rate $\eta = \frac{1}{\mu(t+\gamma)}$ and $\gamma = \frac{4L}{\mu}$, after $T$ iterations of FL with partial device participation, we have the convergence as:*

$$\mathbb{E}[F(\mathbf{w}^{(T)})] - F^* \leq \frac{1}{(T+\gamma)} \left[ \frac{4L(32\tau^2 G^2 + \sigma^2/m)}{3\mu^2\overline{\rho}} \right.$$
$$\left. + \frac{8L^2\Gamma}{\mu^2} + \frac{L\gamma\|\mathbf{w}^{(0)} - \mathbf{w}^*\|^2}{2} \right] + \frac{8L\Gamma}{3\mu}(\frac{\widetilde{\rho}}{\overline{\rho}} - 1). \tag{23}$$

**Theorem 4.2 (Convergence of AFFL).** *Under Assumptions 4.1 to 4.4, for learning rate $\eta = \frac{1}{\mu(t+\gamma)}$ and $\gamma = \frac{4L}{\mu}$, after $T$ iterations, we have the convergence as:*

$$\mathbb{E}[F(\mathbf{w}^{(T)})] - F^* \leq \frac{1}{(T+\gamma)} \left[ \frac{32L}{33\mu^2}(\widetilde{\sigma}^2 + \zeta_T^2) + \right.$$
$$\left. \frac{8L^2\Gamma}{\mu^2} + \frac{L\gamma\|\mathbf{w}^{(0)} - \mathbf{w}^*\|^2}{2} \right], \tag{24}$$

*where $\zeta_T^2$ is upper bound of bias introduced by data distillation, and $\widetilde{\sigma}^2$ is bounded variance of gradients on server.*

We respectively denote $A_{\text{AFFL}}$, $A_{\text{FL}}$, and $\mathcal{B}$ as:

$$A_{\text{AFFL}} = \frac{32L}{33\mu^2}(\widetilde{\sigma}^2 + \zeta_T^2), \quad A_{\text{FL}} = \frac{4L(32\tau^2 G^2 + \sigma^2/m)}{3\mu^2\overline{\rho}},$$
$$\mathcal{B} = \frac{8L\Gamma}{3\mu}(\frac{\widetilde{\rho}}{\overline{\rho}} - 1). \tag{25}$$

*Table 1.* The Top-1 accuracy (%) of FL methods with ConvNet.

| Methods | CIFAR-10 | | | CIFAR-100 | | | SVHN | | |
|---|---|---|---|---|---|---|---|---|---|
| | $\alpha = 0.01$ | $\alpha = 0.05$ | $\alpha = 0.1$ | $\alpha = 0.01$ | $\alpha = 0.05$ | $\alpha = 0.1$ | $\alpha = 0.01$ | $\alpha = 0.05$ | $\alpha = 0.1$ |
| FedAvg | $40.23_{\pm1.43}$ | $54.14_{\pm1.38}$ | $60.32_{\pm1.27}$ | $20.14_{\pm2.02}$ | $22.37_{\pm1.94}$ | $25.92_{\pm1.83}$ | $58.65_{\pm0.96}$ | $78.26_{\pm0.77}$ | $80.72_{\pm0.61}$ |
| FedNova | $45.02_{\pm1.21}$ | $53.23_{\pm1.15}$ | $63.15_{\pm1.04}$ | $20.16_{\pm1.73}$ | $22.41_{\pm1.89}$ | $26.03_{\pm1.68}$ | $63.68_{\pm0.72}$ | $80.41_{\pm0.63}$ | $83.51_{\pm0.64}$ |
| FedProx | $40.72_{\pm1.25}$ | $54.72_{\pm1.18}$ | $59.14_{\pm1.17}$ | $20.37_{\pm1.84}$ | $22.51_{\pm1.78}$ | $25.81_{\pm1.66}$ | $58.71_{\pm0.79}$ | $79.24_{\pm0.86}$ | $80.93_{\pm0.57}$ |
| SCAFFOLD | $34.75_{\pm1.33}$ | $51.53_{\pm1.37}$ | $64.47_{\pm1.22}$ | $23.68_{\pm1.79}$ | $25.17_{\pm1.82}$ | $27.73_{\pm1.53}$ | $66.43_{\pm0.92}$ | $77.65_{\pm0.79}$ | $82.99_{\pm0.64}$ |
| FedConcat | $30.52_{\pm1.39}$ | $35.91_{\pm1.26}$ | $46.26_{\pm1.25}$ | $18.65_{\pm1.96}$ | $21.43_{\pm1.71}$ | $21.31_{\pm1.62}$ | $57.41_{\pm0.85}$ | $73.59_{\pm0.68}$ | $77.03_{\pm0.53}$ |
| FedDM | $61.27_{\pm0.86}$ | $63.15_{\pm0.67}$ | $63.82_{\pm0.74}$ | $31.24_{\pm1.01}$ | $32.58_{\pm0.83}$ | $34.18_{\pm0.84}$ | $80.49_{\pm0.58}$ | $82.31_{\pm0.57}$ | $83.74_{\pm0.46}$ |
| FedAF | $63.85_{\pm0.92}$ | $66.19_{\pm0.64}$ | $67.38_{\pm0.65}$ | $34.53_{\pm0.98}$ | $35.02_{\pm0.99}$ | $37.20_{\pm0.91}$ | $81.49_{\pm0.56}$ | $83.53_{\pm0.45}$ | $84.32_{\pm0.42}$ |
| **FedUSD(ours)** | $\mathbf{67.67_{\pm0.79}}$ | $\mathbf{71.05_{\pm0.77}}$ | $\mathbf{74.73_{\pm0.54}}$ | $\mathbf{44.58_{\pm0.70}}$ | $\mathbf{45.89_{\pm0.89}}$ | $\mathbf{47.17_{\pm0.74}}$ | $\mathbf{88.23_{\pm0.53}}$ | $\mathbf{88.96_{\pm0.31}}$ | $\mathbf{89.96_{\pm0.37}}$ |

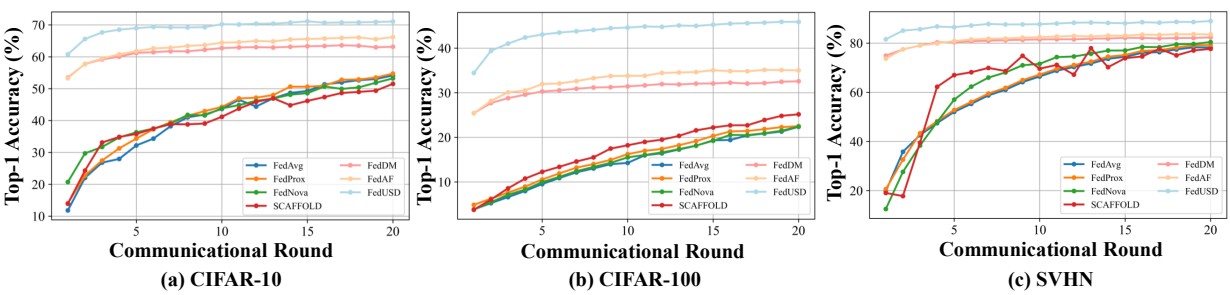

*Figure 4.* Convergence performance of different FL methods with ConvNet, when $\alpha = 0.05$.

If $A_{\text{AFFL}} > A_{\text{FL}}$, AFFL will still outperform after sufficient rounds, and required number of rounds is:

$$T^* = \lceil \frac{A_{\text{AFFL}} - A_{\text{FL}}}{\mathcal{B}} - \gamma \rceil. \tag{26}$$

If $A_{\text{AFFL}} < A_{\text{FL}}$, AFFL is superior starting from first round. It demonstrates the superior convergence performance of AFFL and reveals that the most critical factor affecting its convergence is distillation error $\zeta_T^2$. According to Theorem 3.1, our proposed FedUSD achieves minimum distillation error, thereby enabling it to attain SOTA performance.

## 5. Experiments

### 5.1. Experiments Settings

**Architectures and Datasets**. Following previous studies (Luo et al., 2023; Hao et al., 2025; Xiong et al., 2023; Wang et al., 2024c), we compare the performance of methods on CIFAR-10, CIFAR-100, SVHN (Sermanet et al., 2012) and TinyImageNet. We apply joint learning on 10 clients, the updates will be conducted across whole 10 clients, and the number of synthetic IPC is set to 10 for CIFAR-10 and SVHN, 5 for CIFAR-100 and TinyImageNet, all synthetic data are initialized by sampling from the real data. We use ConvNet (Zhao et al., 2021), ResNet-18 (He et al., 2016), ViT-Tiny-Patch16-224 (ViT-Tiny-16) for CIFAR-10, and ConvNet for other datasets.

**Baseline**. To demonstrate the superiority of our proposed FedUSD, we compare it against several existing Aggregation-Based methods: FedAvg (McMahan et al., 2017), FedProx (Li et al., 2020b), FedNova (Wang et al., 2020), FedConcat (Diao et al., 2024), SCAFFOLD (Karimireddy et al., 2020), and Aggregation-Free methods: FedDM (Xiong et al., 2023), FedAF (Wang et al., 2024c).

**Implementation Details**. Following common settings, for Non-IID dataset partitioning, we use Dirichlet-$\alpha$ sampling (Hsu et al., 2019) where $\alpha$ measures the heterogeneity. The smaller the $\alpha$, the more imbalanced the data distribution. For all experiences, we select the $\alpha$ from $\{0.01, 0.05, 0.1\}$. For synthetic data optimization, we sample 128 real data each epoch, set learning rate to 1, both hyperparameters $\lambda_1$ and $\lambda_2$ to 0.01, which will be discussed in Sec 5.3. To achieve the best trade-off between performance and efficiency, we set SEA factor $l$ to 2. For global model training on server, we uses a cosine decay learning rate with an initial value 0.001, and synthetic dataset participates in training for 3 rounds ($\mathbb{T} = 3$). For training epoch, communication round $T$ is set to 20, synthetic data optimization round $E$ is set to 1000 on each client, and the global model training epoch $R$ is set to 300 on the sever. The mean values for the average accuracy over last 50 epoch of global model training stage on server are calculated from 10 experiments using 5 different random seeds and can be found in Table 1, 2 and figure 4, 5, the best results are bolded.

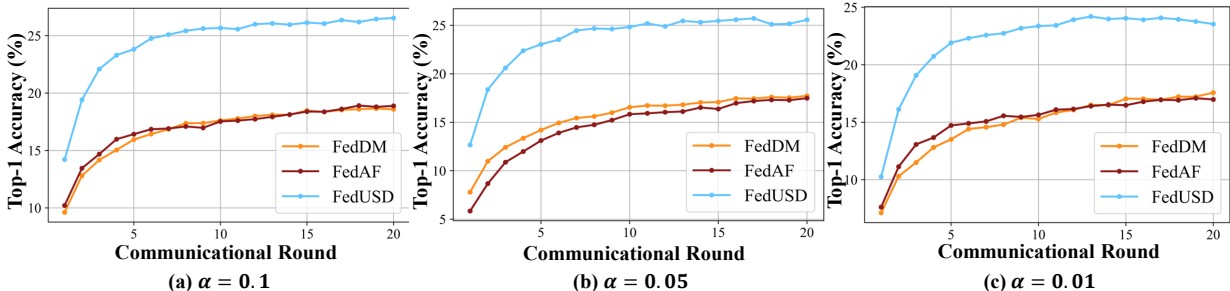

*Figure 5.* Convergence performance of AFFL methods on TinyImageNet with ConvNet.

*Table 2.* The Top-1 accuracy (%) of AFFL methods on CIFAR-10.

| Methods | ConvNet | | | ViT-Tiny-16 | | |
|---|---|---|---|---|---|---|
| | 0.01 | 0.05 | 0.1 | 0.01 | 0.05 | 0.1 |
| FedDM | 61.27 | 63.15 | 63.82 | 38.04 | 43.93 | 45.33 |
| FedAF | 63.85 | 66.19 | 67.38 | 36.71 | 45.33 | 47.81 |
| **FedUSD** | **67.67** | **71.05** | **74.73** | **45.41** | **53.93** | **57.07** |

*Table 3.* Communication costs of FedUSD and two representative methods using ConvNet under $\alpha = 0.01$.

| Dataset | FedAvg | FedDM | FedUSD |
|---|---|---|---|
| CIFAR-10 | 256.0MB | 35.2MB | 35.2MB |
| CIFAR-100 | 403.5MB | 326.0MB | 326.0MB |
| SVHN | 256.0MB | 37.5MB | 37.5MB |

*Table 4.* Communication costs of FedUSD and two representative methods using ResNet-18 under $\alpha = 0.01$.

| Dataset | FedAvg | FedDM | FedUSD |
|---|---|---|---|
| CIFAR-10 | 8.7GB | 35.2MB | 35.2MB |
| CIFAR-100 | 8.8GB | 326.0MB | 326.0MB |
| SVHN | 8.7GB | 37.5MB | 37.5MB |

networks. Since the communication costs of Aggregation-Based FL are similar, we use FedAvg as a representation. To ensure fairness, all architectures and image sizes are uniformly measured in float32 format. The tables summarize the cumulative communication costs of 20 rounds with 10 clients for each method. Aggregation-based FL incurs overhead mainly from uploading local parameters, which becomes costly and inefficient with large models. In contrast, AFFL transmits small synthetic data rather than larger model weights, sharply reducing communication costs and making it more suitable for training large models in FL. Furthermore, the experiment results demonstrate that FedUSD achieves great performance improvements in the global model without incurring any extra communication cost.

### 5.2. Performance Analysis

**Overall Performace.** As shown in Table 1, Figure 4 and Figure 5, under different conditions, FedUSD achieves impressive performance compared to all methods. For example, when $\alpha = 0.1$ on CIFAR-10, FedUSD outperforming other methods by 7.35%-28.47%. Meanwhile, FedUSD shows the fastest convergence reaching the final performance of other methods within just one communication round. As shown in Table 5, even using only $\mathcal{L}_{USD}$, FedUSD outperforms other methods, demonstrating the superiority of $\mathcal{L}_{USD}$. It empirically shows that HOB has a stronger capability than the mean in extracting principal components.

To further investigate the performance of different AFFL methods, we conduct experiments on CIFAR-10 with different models. As shown in Table 2, FedUSD achieves superior performance across various models. For example, when $\alpha = 0.1$ on ViT-Tiny-16, FedUSD outperforming other methods by 9.26%-11.74%.

**Communication Costs.** As shown in Table 3 and 4, we compare the communication costs of FedUSD with two representative FL method across 3 datasets using different

### 5.3. Ablation Study

**Implementation Details.** We design three sets of experiments to investigate the impact of IPC, loss function settings, and each module under the performance of FedUSD. To investigate the module impact, we first design a baseline (BL) where each client randomly samples data and uploads it for server-side training without dataset distillation. Based on BL, we then incrementally add USD optimization and the SEA module to observe their effects for performance on CIFAR-10 and CIFAR-100. For IPC impact, we train ConvNet on CIFAR-10 under 7 IPC settings and observe the results to choose the optimal IPC. To analyze the parameter sensitivity, we perform a grid search over $\lambda_1, \lambda_2 \in \{10^{-4}, 10^{-3}, 10^{-2}, 0.1, 1\}$ by training ConvNet on CIFAR-10 with $\alpha = 0.1$. The mean values for the average accuracy over last 50 epoch of each global model training stage on server are calculated from 10 experiments

*Table 5.* The Top-1 accuracy (%) of methods with different components on CIFAR-10 and CIFAR-100 under 3 heterogeneity levels.

| Components | CIFAR-10 | | | CIFAR-100 | | |
|---|---|---|---|---|---|---|
| | 0.01 | 0.05 | 0.1 | 0.01 | 0.05 | 0.1 |
| Baseline | 56.30 | 60.03 | 62.15 | 28.61 | 31.27 | 33.24 |
| +SEA | 64.50 | 69.81 | 72.50 | 37.78 | 41.12 | 43.47 |
| +USD Loss | 65.41 | 68.13 | 69.61 | 35.57 | 37.59 | 39.52 |
| **FedUSD** | **67.67** | **71.05** | **74.73** | **44.58** | **45.89** | **47.17** |

*Table 6.* The Top-1 Accuracy (%) of FedUSD with different IPC values under three levels of heterogeneity.

| Coeffecient | $\alpha = 0.01$ | $\alpha = 0.05$ | $\alpha = 0.1$ |
|---|---|---|---|
| 1 | 48.14 | 48.59 | 48.14 |
| 5 | 58.88 | 61.13 | 68.61 |
| 10 | 67.67 | 71.05 | 74.73 |
| 20 | 73.59 | 75.14 | 78.23 |
| 30 | 76.41 | 77.69 | 79.46 |
| 40 | 79.36 | 79.09 | 80.66 |
| 50 | 79.24 | 80.08 | 81.14 |

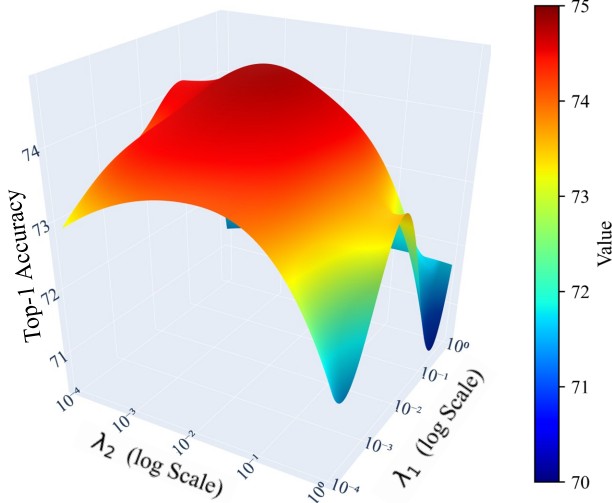

*Figure 6.* The Top-1 Accuracy (%) grid of FedUSD with changing hypeparameters $\lambda_1$ and $\lambda_2$ on CIFAR-10.

$\lambda_1 = 0.01$, $\lambda_2 = 0.01$, it highlights the parameter region associated with strong global model performance. Therefore, we set $\lambda_1$ and $\lambda_2$ to 0.01 in all experiments.

using 5 different random seeds and can be found in Table 5, 6 and Figure 6, the best results are bolded.

**Impact of Core Components.** As shown in Table 5, across all datasets, the removal of either USD Loss or the SEA results in a noticeable drop in accuracy, demonstrating the critical role these two components play in the effectiveness of the FedUSD framework. For example, when training ConvNet on CIFAR-10 with $\alpha = 0.1$, accuracy decreases by 5.12% without USD Loss and 2.23% without SEA. Removing both leads to a decrease of 12.58%. Simikar trends are also observed in other experiments.

**Impact of IPC.** As IPC increases, performance improves (Table 6) which demonstrates the effectiveness of synthetic data optimized by FedUSD. Nonetheless, the improvement becomes marginal when IPC>10, and the performance nearly converges when IPC reaches 30. Thus, to balance communication efficiency and overall performance, we set IPC value as 10 for CIFAR-10. We adopt the same procedure to set IPC for other datasets and methods, aiming to achieve stable performance gains across diverse settings while balancing communication and computation overhead.

**Hyperparameter Sensitivity.** FedUSD matches the HOB and variance between synthetic and real data. To determine the optimal configuration that regulates the contributions of class-level principal and residual components, we conduct a grid search on the CIFAR-10 dataset with Dir($\alpha$)=0.1 to obtain the best-performing parameter set. As shown in Figure 6, the deep red area is notably concentrated around

## 6. Limitation

This work mainly validates the core effectiveness of the proposed method under a unimodal setting. In future work, we will further investigate its applicability to cross-modal tasks, where heterogeneous feature spaces and modality gaps may introduce additional challenges. Extending the proposed method to such scenarios will help evaluate its broader generalization ability.

## 7. Conclusion

In this paper, we propose FedUSD as an AFFL method. By collaboratively matching HOB and variance, we optimize the synthetic data to achieve unbiased synthetic data optimization. Sufficient theoretical proof has verified that FedUSD can generate unbiased synthetic data, and extensive experiments have demonstrated its superior over other existing methods in FL. Additionally, we conduct ablation studies to verify the effectiveness of each module in FedUSD and investigate its hyperparameter settings. These findings reveal the potential of FedUSD to significantly improve the global model performance without introducing any additional communicational cost in FL.

## Acknowledgement

This work was supported in part by the National Natural Science Foundation of China (Grants 62322117, 62531018, 62425109, 62371365, and U24B20136); in part

by the Fundamental and Interdisciplinary Disciplines Breakthrough Plan of the Ministry of Education of China (Grant JYB2025XDXM105); and in part by Fundamental Research Funds for the Central Universities and Innovation Fund of Xidian University (YJSJ26022).

## Impact Statement

This paper presents work whose goal is to advance the field of Machine Learning. There are many potential societal consequences of our work, none which we feel must be specifically highlighted here.

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

# A. Preliminaries for Proof

## A.1. Notation

Necessary notations are introduced as follows.

- $\mathbf{Z}_{ik}$: Hidden representation set of dataset $D_{ik}$ on client $i$ for class $k$.

- $\mathbf{z}_{ij,k}$: Hidden representation j in $\mathbf{Z}_{ik}$.

- $g_{ik}$: The representation of principal component on client $i$ for class $k$. (mean $\mu_{ik}$, HOB $\mathbf{v}_{ik}$.)

- $(\sigma_{ik}^m)^2$: The energy of the $m$-th orthogonal basis.

- $\mathcal{N} = \text{rank}(\mathbf{Z}_{ik})$.

- $n_{ik} = |\mathbf{Z}_{ik}|$.

- $\beta_{ik} = [\beta_{ik}^1, ..., \beta_{ik}^{n_{ik}}]^\top$: Projection of each hidden representation onto $g_{ik}$.

- $\mathbf{w}_i^{(t)}$: Local weight vector on client $i$ at epoch $t$.

- $\mathbf{g}_i(\mathbf{w}_i^{(t)}, \xi_i^{(t)}) = \nabla F_i(\mathbf{w}_i^{(t)}, \xi_i^{(t)})$, $\xi_i^{(t)}$ is mini-batch uniformly sampled at random from $D_i$.

- $\mathbf{w}^{(t)} = \frac{1}{m} \sum_{i \in \mathcal{S}} \mathbf{w}_i^{(t)}$

- $\zeta_t$: Upper bound on the gradient deviation between the real data and the synthetic data at round $t$.

- $\tilde{\sigma}^2$: Upper bound on the server-side mini-batch gradient variance of the synthetic data.

# B. Proof of Theorem 3.1

## B.1. Minimal Reconstruction Error of HOB

We formulate the principal component representation as a rank-1 reconstruction problem, and denoted as:

$$\min_{g_{ik}, \beta_{ik}^1, ..., \beta_{ik}^{n_{ik}}} \mathcal{L}_{pc}(g_{ik}, \beta_{ik}) := \|\mathbf{Z}_{ik} - \beta_{ik} g_{ik}^\top\|_F^2. \tag{27}$$

Then optimize $\beta_{ik}$ for fixed $g_{ik}$:

$$\mathcal{L}_{pc}(g_{ik}, \beta_{ik}) = \text{Tr}[(\mathbf{Z}_{ik} - \beta_{ik} g_{ik}^\top)(\mathbf{Z}_{ik} - \beta_{ik} g_{ik}^\top)^\top]$$
$$= \|\mathbf{Z}_{ik}\|_F^2 - 2\beta_{ik}^\top \mathbf{Z}_{ik} g_{ik} + \|\beta_{ik}\|_2^2. \tag{28}$$

Setting the gradient to 0 yields:

$$\frac{\partial \mathcal{L}_{pc}}{\partial \beta_{ik}} = -2\mathbf{Z}_{ik} g_{ik} + 2\beta_{ik} = 0. \tag{29}$$

So, we can get a solution:

$$\beta_{ik}^* = \mathbf{Z}_{ik} g_{ik}. \tag{30}$$

Substitute it back to Equation 28:

$$\mathcal{L}_{pc}(g_{ik}, \beta_{ik}^*) = \|\mathbf{Z}_{ik}\|_F^2 - \|\mathbf{Z}_{ik} g_{ik}\|_2^2, \tag{31}$$

where we can denote $\|\mathbf{Z}_{ik} g_{ik}\|_2^2$ as:

$$\|\mathbf{Z}_{ik} g_{ik}\|_2^2 = g_{ik}^\top \mathbf{Z}_{ik}^\top \mathbf{Z}_{ik} g_{ik} = g_{ik}^\top C g_{ik}, \tag{32}$$

where $C = \mathbf{Z}_{ik}^\top \mathbf{Z}_{ik}$, hence the original problem (Equation 27) is equivalent to:

$$\max_{\|g_{ik}\|_2 = 1} g_{ik}^\top C g_{ik}. \tag{33}$$

It is the Rayleigh-quotient problem, its maximum is the largest eigenvalue of $C$, i.e. $(\sigma_{ik}^1)^2$ attain at $g_{ik} = \mathbf{v}_{ik}^\top$. Therefore, we can get the conclusion that the optimal principal component direction is:

$$g_{ik}^* = \mathbf{v}_{ik}^\top, \beta_{ik}^{j*} = < \mathbf{z}_{ij,k}, \mathbf{v}_{ik} >, \tag{34}$$

and the minimal error is:

$$e_{HOB} = \min \mathcal{L}_{pc} = \sum_{m=2}^{\mathcal{N}} (\sigma_{ik}^m)^2. \tag{35}$$

## B.2. Minimal Reconstruction Error of Mean

To fairest compare with SVD, we define the mean as:

$$\mu_{ik} = \frac{\frac{1}{n_{ik}}\mathbf{1}^\top \mathbf{Z}_{ik}}{\|\frac{1}{n_{ik}}\mathbf{1}^\top \mathbf{Z}_{ik}\|_2}, \mathbf{1} = [1, ...1]^\top \in \mathbb{R}^{n_{ik} \times 1}. \tag{36}$$

We fix all coefficients to 1, then the reconstruction matrix is $\mathbf{1}\mu_{ik}$, and the minimal error is:

$$\begin{aligned} e_{mean} &= \|\mathbf{Z}_{ik} - \mathbf{1}\mu_{ik}\|_F^2 \\ &= \|\mathbf{Z}_{ik}\|_F^2 + \|\mathbf{1}\mu_{ik}\|_F^2 - 2\mathrm{Tr}(\mathbf{Z}_{ik}(\mathbf{1}\mu_{ik})^\top). \end{aligned} \tag{37}$$

## B.3. Comparison of $e_{\text{HOB}}$ and $e_{\text{mean}}$

By computing Equation 37, we can get:

$$\begin{aligned} \mathrm{Tr}(\mathbf{Z}_{ik}(\mathbf{1}\mu_{ik})^\top) &= \mathrm{Tr}(\mathbf{Z}_{ik}g_{ik}\mathbf{1}^\top) \\ &= \mathbf{1}^\top \mathbf{Z}_{ik}g_{ik} \\ &= \mathbf{1}^\top(\mathbf{Z}_{ik}g_{ik}). \end{aligned} \tag{38}$$

We denote $\mathbf{M} = \mathbf{Z}_{ik}g_{ik}$, $\|\mathbf{M}\|_2 = \|\mathbf{Z}_{ik}g_{ik}\|_2$, then, substitute Equations 38 into Equation 37, we obtain:

$$e_{mean} = \|\mathbf{Z}_{ik}\|_F^2 + n_{ik} - 2\mathbf{1}^\top \mathbf{M}. \tag{39}$$

Apply Cauchy–Schwarz to $\mathbf{1}^\top \mathbf{M}$:

$$|\mathbf{1}^\top \mathbf{M}| \le \|\mathbf{1}\|_2 \|\mathbf{M}\|_2 = \sqrt{n_{ik}}\|\mathbf{M}\|_2. \tag{40}$$

So, we can get a lower bound for $e_{mean}$:

$$e_{mean} \ge \|\mathbf{Z}_{ik}\|_F^2 + n_{ik} - 2\sqrt{n_{ik}}\|\mathbf{M}\|_2. \tag{41}$$

Let $a = \|\mathbf{M}\|_2 \ge 0$:

$$\begin{aligned} e_{mean} &\ge \|\mathbf{Z}_{ik}\|_F^2 - a^2 + (a - \sqrt{n_{ik}})^2 \\ &\ge \|\mathbf{Z}_{ik}\|_F^2 - a^2. \end{aligned} \tag{42}$$

So, we can get:

$$e_{mean} \ge \|\mathbf{Z}_{ik}\|_F^2 - \|\mathbf{Z}_{ik}g_{ik}\|_2^2. \tag{43}$$

Because $g_{ik} = \mu_{ik}^\top$ satisfies $\|g_{ik}\|_2 = 1$,

$$\|\mathbf{Z}_{ik}g_{ik}\|_2^2 = g_{ik}^\top \mathbf{Z}_{ik}^\top \mathbf{Z}_{ik}g_{ik}. \tag{44}$$

The symmetric positive-semidefinite matrix $C = \mathbf{Z}_{ik}^\top \mathbf{Z}_{ik}$ has maximum eigenvalue $\sigma_1^2$, the Rayleigh-quotient theorem shows:

$$g_{ik}^\top C g_{ik} \le (\sigma_{ik}^1)^2, \tag{45}$$

with equality only when $g_{ik} = \mathbf{v}_{ik}^\top$, so:

$$\|\mathbf{Z}_{ik}g_{ik}\|_2^2 \leq (\sigma_{ik}^1)^2. \tag{46}$$

Substituting Equation 46 into Equation 43 yields:

$$e_{mean} \geq \|\mathbf{Z}_{ik}\|_F^2 - (\sigma_{ik}^1)^2, \tag{47}$$

$$e_{mean} \geq \sum_{m=2}^{\mathcal{N}} (\sigma_{ik}^m)^2 + ((\sigma_{ik}^1)^2 - \|\mathbf{Z}_{ik}\mu_{ik}^\top\|_2^2). \tag{48}$$

According to the Assumption 3.1 in section "Methodology", we can get $(\sigma_{ik}^1)^2 - \|\mathbf{Z}_{ik}\mu_{ik}^\top\|_2^2 \geq 0$, so we get the conclusion:

$$e_{mean} \geq e_{HOB}. \tag{49}$$

Assume that all hidden representations satisfy:

$$\mathbf{z}_{ij,k} = \gamma_j \mathbf{v}_{ik}^\top + \Xi_j^\top, < \Xi_j, \mathbf{v}_{ik} >= 0. \tag{50}$$

When $\mathbf{v}_{ik}\|\mu_{ik}$, substituting Equation 50 into Equation 48 yields:

$$e_{mean} = \sum_{m=2}^{\mathcal{N}} (\sigma_{ik}^m)^2 + \text{Var}(\gamma_j)\|\mathbf{v}_{ik}^\top\|^2. \tag{51}$$

So, equality is achieved if and only if $\mathbf{v}_{ik}\|\mu_{ik}$ and all hidden representations have approximately same magnitude.

## C. Preliminaries for Proof of Theorem 4.1 and 4.2

We present the preliminary lemmas used for proof of Theorem 4.1 and 4.2. We will denote the expectation over the sampling random source $\mathcal{S}(t)$ as $\mathbb{E}_{\mathcal{S}(t)}$ and the expectation over all the random sources as $\mathbb{E}$.

**Lemma C.1.** *Suppose $F_i$ is L-smooth with global minimum at $\mathbf{w}_i^*$, then for any $\mathbf{w}_i$ in the domain of $F_i$, we have that*

$$\|\nabla F_i(\mathbf{w}_i)\|^2 \leq 2L(F_i(\mathbf{w}_i) - F_i(\mathbf{w}_i^*)). \tag{52}$$

*Proof.*

$$F_i(\mathbf{w}_i) - F_i(\mathbf{w}_i^*) - < \nabla F_i(\mathbf{w}_i^*), \mathbf{w}_i - \mathbf{w}_i^* >\geq \frac{1}{2L}\|\nabla F_i(\mathbf{w}_i) - \nabla F_i(\mathbf{w}_i^*)\|^2. \tag{53}$$

$$F_i(\mathbf{w}_i) - F_i(\mathbf{w}_i^*) \geq \frac{1}{2L}\|\nabla F_i(\mathbf{w}_i)\|^2. \tag{54}$$

**Lemma C.2** (Expected average discrepancy between $\mathbf{w}^{(t)}$ and $\mathbf{w}_i^{(t)}$ for $i \in \mathcal{S}(t)$)**.**

$$\frac{1}{m}\mathbb{E}[\sum_{i\in\mathcal{S}(t)} \|\mathbf{w}^{(t)} - \mathbf{w}_i^{(t)}\|^2] \leq 16\eta_t^2\tau^2 G^2. \tag{55}$$

*Proof.*

$$\begin{aligned}
\frac{1}{m}\sum_{i\in\mathcal{S}(t)} \|\mathbf{w}^{(t)} - \mathbf{w}_i^{(t)}\|^2 &= \frac{1}{m}\sum_{i\in\mathcal{S}(t)} \|\frac{1}{m}\sum_{i'\in\mathcal{S}(t)} (\mathbf{w}_{i'}^{(t)} - \mathbf{w}_i^{(t)})\|^2 \\
&\leq \frac{1}{m^2}\sum_{i\in\mathcal{S}(t)}\sum_{i'\in\mathcal{S}(t)} \|(\mathbf{w}_{i'}^{(t)} - \mathbf{w}_i^{(t)})\|^2 \\
&= \frac{1}{m^2}\sum_{\substack{i\neq i' \\ i,i'\in\mathcal{S}(t)}} \|(\mathbf{w}_{i'}^{(t)} - \mathbf{w}_i^{(t)})\|^2.
\end{aligned} \tag{56}$$

Moreover, for any $t$, there is a $t_0$ such that $\mathbf{w}_{i'}^{(t_0)} = \mathbf{w}_i^{(t_0)}$ and $0 \leq t - t_0 < \tau$, because the selected clients are updated with the global model at every $\tau$. Hence, even for an arbitrary $t$, we have the difference $\|\mathbf{w}_{i'}^{(t)} - \mathbf{w}_i^{(t)}\|^2$ and it is upper bounded by $\tau$ updates. With non-increasing $\eta_t$ over $t$ and $\eta_{t_0} \leq 2\eta_t$, Equation 56 can be further bounded as,

$$
\begin{aligned}
\frac{1}{m^2} \sum_{\substack{i \neq i' \\ i, i' \in \mathcal{S}(t)}} \|\mathbf{w}_{i'}^{(t)} - \mathbf{w}_i^{(t)}\|^2 &\leq \frac{1}{m^2} \sum_{\substack{i \neq i' \\ i, i' \in \mathcal{S}(t)}} \| \sum_{j=t_0}^{t_0+\tau-1} \eta_j (\mathbf{g}_{i'}(\mathbf{w}_{i'}^{(j)}, \xi_{i'}^{(j)}) - \mathbf{g}_i(\mathbf{w}_i^{(j)}, \xi_i^{(j)}))\|^2 \\
&\leq \frac{\eta_{t_0}^2 \tau}{m^2} \sum_{\substack{i \neq i' \\ i, i' \in \mathcal{S}(t)}} \sum_{j=t_0}^{t_0+\tau-1} \|\mathbf{g}_{i'}(\mathbf{w}_{i'}^{(j)}, \xi_{i'}^{(j)}) - \mathbf{g}_i(\mathbf{w}_i^{(j)}, \xi_i^{(j)})\|^2 \\
&\leq \frac{\eta_{t_0}^2 \tau}{m^2} \sum_{\substack{i \neq i' \\ i, i' \in \mathcal{S}(t)}} \sum_{j=t_0}^{t_0+\tau-1} \left[ 2\|\mathbf{g}_{i'}(\mathbf{w}_{i'}^{(j)}, \xi_{i'}^{(j)})\|^2 + 2\|\mathbf{g}_i(\mathbf{w}_i^{(j)}, \xi_i^{(j)})\|^2 \right].
\end{aligned}
\tag{57}
$$

By taking expectation over Equation 57, we can get $\mathbb{E}[\frac{1}{m^2} \sum_{\substack{i \neq i' \\ i, i' \in \mathcal{S}(t)}} \|\mathbf{w}_{i'}^{(t)} - \mathbf{w}_i^{(t)}\|^2]$.

$$
\begin{aligned}
\mathbb{E}[\frac{1}{m^2} \sum_{\substack{i \neq i' \\ i, i' \in \mathcal{S}(t)}} \|\mathbf{w}_{i'}^{(t)} - \mathbf{w}_i^{(t)}\|^2] &\leq \frac{2\eta_{t_0}^2 \tau}{m^2} \mathbb{E}[\sum_{\substack{i \neq i' \\ i, i' \in \mathcal{S}(t)}} \sum_{j=t_0}^{t_0+\tau-1} (\|\mathbf{g}_{i'}(\mathbf{w}_{i'}^{(j)}, \xi_{i'}^{(j)})\|^2 + \|\mathbf{g}_i(\mathbf{w}_i^{(j)}, \xi_i^{(j)})\|^2)] \\
&\leq \frac{2\eta_{t_0}^2 \tau}{m^2} \mathbb{E}_{\mathcal{S}(t)}[\sum_{\substack{i \neq i' \\ i, i' \in \mathcal{S}(t)}} \sum_{j=t_0}^{t_0+\tau-1} 2G^2] = \frac{2\eta_{t_0}^2 \tau}{m^2} \mathbb{E}_{\mathcal{S}(t)}[\sum_{\substack{i \neq i' \\ i, i' \in \mathcal{S}(t)}} 2\tau G^2] \\
&\leq \frac{16\eta_t^2 (m-1)\tau^2 G^2}{m} \leq 16\eta_t^2 \tau^2 G^2.
\end{aligned}
\tag{58}
$$

**Lemma C.3** (Upper bound for expectation over $\|\mathbf{w}^{(t)} - \mathbf{w}^*\|$). *By using $\mathbb{E}[\cdot]$, we have the upper bound of the total expectation over all random sources as:*

$$
\mathbb{E}[\|\mathbf{w}^{(t)} - \mathbf{w}^*\|^2] \leq \frac{1}{m} \mathbb{E}[\sum_{i \in \mathcal{S}(t)} \|\mathbf{w}_i^{(t)} - \mathbf{w}^*\|^2].
\tag{59}
$$

*Proof.*

$$
\mathbb{E}[\|\mathbf{w}^{(t)} - \mathbf{w}^*\|^2] = \mathbb{E}[\|\frac{1}{m} \sum_{i \in \mathcal{S}(t)} \mathbf{w}_i^{(t)} - \mathbf{w}^*\|^2] = \mathbb{E}[\|\frac{1}{m} \sum_{i \in \mathcal{S}(t)} (\mathbf{w}_i^{(t)} - \mathbf{w}^*)\|^2] \leq \frac{1}{m} \mathbb{E}[\sum_{i \in \mathcal{S}(t)} \|\mathbf{w}_i^{(t)} - \mathbf{w}^*\|^2].
\tag{60}
$$

# D. Proof of Theorem 4.1

With $\overline{\mathbf{g}}^{(t)} = \frac{1}{m} \sum_{i \in \mathcal{S}(t)} \mathbf{g}_i(\mathbf{w}_i^{(t)}, \xi_i^{(t)})$, we have that:

$$
\begin{aligned}
\|\mathbf{w}^{(t+1)} - \mathbf{w}^*\|^2 &= \|\mathbf{w}^{(t)} - \eta_t \overline{\mathbf{g}}^{(t)} - \mathbf{w}^*\|^2 \\
&= \|\mathbf{w}^{(t)} - \eta_t \overline{\mathbf{g}}^{(t)} - \mathbf{w}^* - \frac{\eta_t}{m} \sum_{i \in \mathcal{S}(t)} \nabla F_i(\mathbf{w}_i^{(t)}) + \frac{\eta_t}{m} \sum_{i \in \mathcal{S}(t)} \nabla F_i(\mathbf{w}_i^{(t)})\|^2 \\
&= \|\mathbf{w}^{(t)} - \mathbf{w}^* - \frac{\eta_t}{m} \sum_{i \in \mathcal{S}(t)} \nabla F_i(\mathbf{w}_i^{(t)})\|^2 + \eta_t^2 \|\frac{1}{m} \sum_{i \in \mathcal{S}(t)} \nabla F_i(\mathbf{w}_i^{(t)}) - \overline{\mathbf{g}}^{(t)}\|^2 \\
&\quad + 2\eta_t < \mathbf{w}^{(t)} - \mathbf{w}^* - \frac{\eta_t}{m} \sum_{i \in \mathcal{S}(t)} \nabla F_i(\mathbf{w}_i^{(t)}), \frac{1}{m} \sum_{i \in \mathcal{S}(t)} \nabla F_i(\mathbf{w}_i^{(t)}) - \overline{\mathbf{g}}^{(t)} > \\
&= \|\mathbf{w}^{(t)} - \mathbf{w}^*\|^2 - 2\eta_t < \mathbf{w}^{(t)} - \mathbf{w}^*, \frac{1}{m} \sum_{i \in \mathcal{S}(t)} \nabla F_i(\mathbf{w}_i^{(t)}) > \\
&\quad + 2\eta_t < \mathbf{w}^{(t)} - \mathbf{w}^* - \frac{\eta_t}{m} \sum_{i \in \mathcal{S}(t)} \nabla F_i(\mathbf{w}_i^{(t)}), \frac{1}{m} \sum_{i \in \mathcal{S}(t)} \nabla F_i(\mathbf{w}_i^{(t)}) - \overline{\mathbf{g}}^{(t)} > \\
&\quad + \eta_t^2 \|\frac{1}{m} \sum_{i \in \mathcal{S}(t)} \nabla F_i(\mathbf{w}_i^{(t)})\|^2 + \eta_t^2 \|\frac{1}{m} \sum_{i \in \mathcal{S}(t)} \nabla F_i(\mathbf{w}_i^{(t)}) - \overline{\mathbf{g}}^{(t)}\|^2.
\end{aligned}
\tag{61}
$$

Then we define:

$$
A_1 = -2\eta_t < \mathbf{w}^{(t)} - \mathbf{w}^*, \frac{1}{m} \sum_{i \in \mathcal{S}(t)} \nabla F_i(\mathbf{w}_i^{(t)}) >,
\tag{62}
$$

$$
A_2 = 2\eta_t < \mathbf{w}^{(t)} - \mathbf{w}^* - \frac{\eta_t}{m} \sum_{i \in \mathcal{S}(t)} \nabla F_i(\mathbf{w}_i^{(t)}), \frac{1}{m} \sum_{i \in \mathcal{S}(t)} \nabla F_i(\mathbf{w}_i^{(t)}) - \overline{\mathbf{g}}^{(t)} >,
\tag{63}
$$

$$
A_3 = \eta_t^2 \|\frac{1}{m} \sum_{i \in \mathcal{S}(t)} \nabla F_i(\mathbf{w}_i^{(t)})\|^2,
\tag{64}
$$

$$
A_4 = \eta_t^2 \|\frac{1}{m} \sum_{i \in \mathcal{S}(t)} \nabla F_i(\mathbf{w}_i^{(t)}) - \overline{\mathbf{g}}^{(t)}\|^2.
\tag{65}
$$

First, to obtain the upper bound of $A_1$ (Equation 62):

$$
\begin{aligned}
-2\eta_t < \mathbf{w}^{(t)} - \mathbf{w}^*, \frac{1}{m} \sum_{i \in \mathcal{S}(t)} \nabla F_i(\mathbf{w}_i^{(t)}) > &= -\frac{2\eta_t}{m} \sum_{i \in \mathcal{S}(t)} < \mathbf{w}^{(t)} - \mathbf{w}^*, \nabla F_i(\mathbf{w}_i^{(t)}) > \\
&= -\frac{2\eta_t}{m} \sum_{i \in \mathcal{S}(t)} < \mathbf{w}^{(t)} - \mathbf{w}_i^{(t)}, \nabla F_i(\mathbf{w}_i^{(t)}) > \\
&\quad - \frac{2\eta_t}{m} \sum_{i \in \mathcal{S}(t)} < \mathbf{w}_i^{(t)} - \mathbf{w}^*, \nabla F_i(\mathbf{w}_i^{(t)}) > .
\end{aligned}
\tag{66}
$$

By using the Cauchy inequality and AM-GM inequality, we can derive a new inequality:

$$
\begin{aligned}
&-\frac{2\eta_t}{m} \sum_{i \in \mathcal{S}(t)} < \mathbf{w}^{(t)} - \mathbf{w}_i^{(t)}, \nabla F_i(\mathbf{w}_i^{(t)}) > -\frac{2\eta_t}{m} \sum_{i \in \mathcal{S}(t)} < \mathbf{w}_i^{(t)} - \mathbf{w}^*, \nabla F_i(\mathbf{w}_i^{(t)}) > \\
&\leq \frac{\eta_t}{m} \sum_{i \in \mathcal{S}(t)} (\frac{1}{\eta_t} \|\mathbf{w}^{(t)} - \mathbf{w}_i^{(t)}\|^2 + \eta_t \|\nabla F_i(\mathbf{w}_i^{(t)})\|^2) - \frac{2\eta_t}{m} \sum_{i \in \mathcal{S}(t)} < \mathbf{w}_i^{(t)} - \mathbf{w}^*, \nabla F_i(\mathbf{w}_i^{(t)}) > \\
&= \frac{1}{m} \sum_{i \in \mathcal{S}(t)} \|\mathbf{w}^{(t)} - \mathbf{w}_i^{(t)}\|^2 + \frac{\eta_t^2}{m} \sum_{i \in \mathcal{S}(t)} \|\nabla F_i(\mathbf{w}_i^{(t)})\|^2 - \frac{2\eta_t}{m} \sum_{i \in \mathcal{S}(t)} < \mathbf{w}_i^{(t)} - \mathbf{w}^*, \nabla F_i(\mathbf{w}_i^{(t)}) > .
\end{aligned}
\tag{67}
$$

Due to the Lemma C.1, we can get a new inequality:

$$\frac{1}{m}\sum_{i\in\mathcal{S}(t)}\|\mathbf{w}^{(t)}-\mathbf{w}_i^{(t)}\|^2+\frac{\eta_t^2}{m}\sum_{i\in\mathcal{S}(t)}\|\nabla F_i(\mathbf{w}_i^{(t)})\|^2-\frac{2\eta_t}{m}\sum_{i\in\mathcal{S}(t)}<\mathbf{w}_i^{(t)}-\mathbf{w}^*,\nabla F_i(\mathbf{w}_i^{(t)})>$$

$$\leq\frac{1}{m}\sum_{i\in\mathcal{S}(t)}\|\mathbf{w}^{(t)}-\mathbf{w}_i^{(t)}\|^2+\frac{2L\eta_t^2}{m}\sum_{i\in\mathcal{S}(t)}(F_i(\mathbf{w}_i^{(t)})-F_i^*)-\frac{2\eta_t}{m}\sum_{i\in\mathcal{S}(t)}<\mathbf{w}_i^{(t)}-\mathbf{w}^*,\nabla F_i(\mathbf{w}_i^{(t)})>.\tag{68}$$

Due to the $\mu$-convexity of $F_i$ (Assumption 4.2):

$$\frac{1}{m}\sum_{i\in\mathcal{S}(t)}\|\mathbf{w}^{(t)}-\mathbf{w}_i^{(t)}\|^2+\frac{2L\eta_t^2}{m}\sum_{i\in\mathcal{S}(t)}(F_i(\mathbf{w}_i^{(t)})-F_i^*)-\frac{2\eta_t}{m}\sum_{i\in\mathcal{S}(t)}<\mathbf{w}_i^{(t)}-\mathbf{w}^*,\nabla F_i(\mathbf{w}_i^{(t)})>$$

$$\leq\frac{1}{m}\sum_{i\in\mathcal{S}(t)}\|\mathbf{w}^{(t)}-\mathbf{w}_i^{(t)}\|^2+\frac{2L\eta_t^2}{m}\sum_{i\in\mathcal{S}(t)}(F_i(\mathbf{w}_i^{(t)})-F_i^*)-\frac{2\eta_t}{m}\sum_{i\in\mathcal{S}(t)}[(F_i(\mathbf{w}_i^{(t)})-F_i(\mathbf{w}^*))+\frac{\mu}{2}\|\mathbf{w}_i^{(t)}-\mathbf{w}^*\|^2].\tag{69}$$

Due to the Lemma C.2:

$$\frac{1}{m}\sum_{i\in\mathcal{S}(t)}\|\mathbf{w}^{(t)}-\mathbf{w}_i^{(t)}\|^2+\frac{2L\eta_t^2}{m}\sum_{i\in\mathcal{S}(t)}(F_i(\mathbf{w}_i^{(t)})-F_i^*)-\frac{2\eta_t}{m}\sum_{i\in\mathcal{S}(t)}[(F_i(\mathbf{w}_i^{(t)})-F_i(\mathbf{w}^*))+\frac{\mu}{2}\|\mathbf{w}_i^{(t)}-\mathbf{w}^*\|^2]$$

$$\leq16\eta_t^2\tau^2G^2-\frac{\eta_t\mu}{m}\sum_{i\in\mathcal{S}(t)}\|\mathbf{w}_i^{(t)}-\mathbf{w}^*\|^2+\frac{2L\eta_t^2}{m}\sum_{i\in\mathcal{S}(t)}(F_i(\mathbf{w}_i^{(t)})-F_i^*)-\frac{2\eta_t}{m}\sum_{i\in\mathcal{S}(t)}(F_i(\mathbf{w}_i^{(t)})-F_i(\mathbf{w}^*)).\tag{70}$$

Next, in expectation, $\mathbb{E}[A_2]=0$ due to the unbiased gradient. Then, we obtain the upper bound of $A_3$ (Equation 64).

$$\eta_t^2\|\frac{1}{m}\sum_{i\in\mathcal{S}(t)}\nabla F_i(\mathbf{w}_i^{(t)})\|^2=\frac{\eta_t^2}{m}\sum_{i\in\mathcal{S}(t)}\|\nabla F_i(\mathbf{w}_i^{(t)})\|^2\leq\frac{2L\eta_t^2}{m}\sum_{i\in\mathcal{S}(t)}(F_i(\mathbf{w}_i^{(t)})-F_i^*).\tag{71}$$

Finally, we can bound $A_4$ by using the bound of variance of stochastic gradients (Assumption 4.3):

$$\mathbb{E}[\eta_t^2\|\frac{1}{m}\sum_{i\in\mathcal{S}(t)}\nabla F_i(\mathbf{w}_i^{(t)})-\overline{\mathbf{g}}^{(t)}\|^2]=\eta_t^2\mathbb{E}[\|\sum_{i\in\mathcal{S}(t)}\frac{1}{m}(\mathbf{g}_i(\mathbf{w}_i^{(t)},\xi_i^{(t)})-\nabla F_i(\mathbf{w}_i^{(t)}))\|^2]$$

$$=\frac{\eta_t^2}{m^2}\mathbb{E}_{\mathcal{S}(t)}[\sum_{i\in\mathcal{S}(t)}\mathbb{E}\|\mathbf{g}_i(\mathbf{w}_i^{(t)},\xi_i^{(t)})-\nabla F_i(\mathbf{w}_i^{(t)})\|^2]$$

$$\leq\frac{\eta_t^2\sigma^2}{m}.\tag{72}$$

Using the upper bounds of $A_1$ (Equation 70), $A_2$, $A_3$ (Equation 71) and $A_4$ (Equation 72), we know the expectation of Equation 61 is bounded as:

$$\mathbb{E}[\|\mathbf{w}^{(t+1)}-\mathbf{w}^*\|^2]\leq\mathbb{E}[\|\mathbf{w}^{(t)}-\mathbf{w}^*\|^2]-\frac{\eta_t\mu}{m}\mathbb{E}[\sum_{i\in\mathcal{S}(t)}\|\mathbf{w}_i^{(t)}-\mathbf{w}^*\|^2]+16\eta_t^2\tau^2G^2+\frac{\eta_t^2\sigma^2}{m}$$

$$+\frac{4L\eta_t^2}{m}\mathbb{E}[\sum_{i\in\mathcal{S}(t)}(F_i(\mathbf{w}_i^{(t)})-F_i^*)]-\frac{2\eta_t}{m}\mathbb{E}[\sum_{i\in\mathcal{S}(t)}(F_i(\mathbf{w}_i^{(t)})-F_i(\mathbf{w}^*))].\tag{73}$$

Due to the Lemma C.3, we can get the bound of equation 73:

$$\mathbb{E}[\|\mathbf{w}^{(t+1)} - \mathbf{w}^*\|^2] \leq (1 - \eta_t\mu)\mathbb{E}[\|\mathbf{w}^{(t)} - \mathbf{w}^*\|^2] + 16\eta_t^2\tau^2 G^2 + \frac{\eta_t^2\sigma^2}{m}$$
$$+ \frac{4L\eta_t^2}{m}\mathbb{E}[\sum_{i \in \mathcal{S}(t)} (F_i(\mathbf{w}_i^{(t)}) - F_i^*)] - \frac{2\eta_t}{m}\mathbb{E}[\sum_{i \in \mathcal{S}(t)} (F_i(\mathbf{w}_i^{(t)}) - F_i(\mathbf{w}^*))]. \tag{74}$$

Then we define:

$$A_5 = \frac{4L\eta_t^2}{m}\mathbb{E}[\sum_{i \in \mathcal{S}(t)} (F_i(\mathbf{w}_i^{(t)}) - F_i^*)] - \frac{2\eta_t}{m}\mathbb{E}[\sum_{i \in \mathcal{S}(t)} (F_i(\mathbf{w}_i^{(t)}) - F_i(\mathbf{w}^*))]. \tag{75}$$

We can represent $A_5$ in a different form as:

$$\mathbb{E}[\frac{4L\eta_t^2}{m}\sum_{i \in \mathcal{S}(t)} (F_i(\mathbf{w}_i^{(t)}) - F_i^*) - \frac{2\eta_t}{m}\sum_{i \in \mathcal{S}(t)} (F_i(\mathbf{w}_i^{(t)}) - F_i(\mathbf{w}^*))]$$
$$= \mathbb{E}[\frac{4L\eta_t^2}{m}\sum_{i \in \mathcal{S}(t)} F_i(\mathbf{w}_i^{(t)}) - \frac{2\eta_t}{m}\sum_{i \in \mathcal{S}(t)} F_i(\mathbf{w}_i^{(t)}) - \frac{2\eta_t}{m}\sum_{i \in \mathcal{S}(t)} (F_i^* - F_i(\mathbf{w}^*)) + \frac{2\eta_t}{m}\sum_{i \in \mathcal{S}(t)} F_i^* - \frac{4L\eta_t^2}{m}\sum_{i \in \mathcal{S}(t)} F_i^*]$$
$$= \mathbb{E}[\frac{2\eta_t(2L\eta_t - 1)}{m}\sum_{i \in \mathcal{S}(t)} (F_i(\mathbf{w}_i^{(t)}) - F_i^*)] + 2\eta_t\mathbb{E}[\frac{1}{m}\sum_{i \in \mathcal{S}(t)} (F_i(\mathbf{w}_i^*) - F_i^*)]. \tag{76}$$

Now, we define:

$$A_6 = \mathbb{E}[\frac{2\eta_t(2L\eta_t - 1)}{m}\sum_{i \in \mathcal{S}(t)} (F_i(\mathbf{w}_i^{(t)}) - F_i^*)]. \tag{77}$$

Now, with $\eta_t < \frac{1}{4L}$ and $v_t = 2\eta_t(1 - 2L\eta_t)$, $A_6$ can be bounded as:

$$-\frac{v_t}{m}\sum_{i \in \mathcal{S}(t)} (F_i(\mathbf{w}_i^{(t)}) - F_i(\mathbf{w}^{(t)}) + F_i(\mathbf{w}^{(t)}) - F_i^*)$$
$$= -\frac{v_t}{m}\sum_{i \in \mathcal{S}(t)} (F_i(\mathbf{w}_i^{(t)}) - F_i(\mathbf{w}^{(t)})) - \frac{v_t}{m}\sum_{i \in \mathcal{S}(t)} (F_i(\mathbf{w}^{(t)}) - F_i^*). \tag{78}$$

Due to the $\mu$-convexity (Assumption 4.2):

$$-\frac{v_t}{m}\sum_{i \in \mathcal{S}(t)} (F_i(\mathbf{w}_i^{(t)}) - F_i(\mathbf{w}^{(t)})) - \frac{v_t}{m}\sum_{i \in \mathcal{S}(t)} (F_i(\mathbf{w}^{(t)}) - F_i^*)$$
$$\leq -\frac{v_t}{m}\sum_{i \in \mathcal{S}(t)} \left[< \nabla F_i(\mathbf{w}^{(t)}), \mathbf{w}_i^{(t)} - \mathbf{w}^{(t)} > + \frac{\mu}{2}\|\mathbf{w}_i^{(t)} - \mathbf{w}^{(t)}\|^2\right] - \frac{v_t}{m}\sum_{i \in \mathcal{S}(t)} (F_i(\mathbf{w}^{(t)}) - F_i^*). \tag{79}$$

Due to the Lemma C.1 and the AM-GM inequality and Cauchy–Schwarz inequality:

$$-\frac{v_t}{m}\sum_{i \in \mathcal{S}(t)} \left[< \nabla F_i(\mathbf{w}^{(t)}), \mathbf{w}_i^{(t)} - \mathbf{w}^{(t)} > + \frac{\mu}{2}\|\mathbf{w}_i^{(t)} - \mathbf{w}^{(t)}\|^2\right] - \frac{v_t}{m}\sum_{i \in \mathcal{S}(t)} (F_i(\mathbf{w}^{(t)}) - F_i^*)$$
$$\leq \frac{v_t}{m}\sum_{i \in \mathcal{S}(t)} \left[\eta_t L(F_i(\mathbf{w}^{(t)}) - F_i^*) + (\frac{1}{2\eta_t} - \frac{\mu}{2})\|\mathbf{w}_i^{(t)} - \mathbf{w}^{(t)}\|^2\right] - \frac{v_t}{m}\sum_{i \in \mathcal{S}(t)} (F_i(\mathbf{w}^{(t)}) - F_i^*)$$
$$= -\frac{v_t}{m}(1 - \eta_t L)\sum_{i \in \mathcal{S}(t)} (F_i(\mathbf{w}^{(t)}) - F_i^*) + (\frac{v_t}{2\eta_t m} - \frac{v_t\mu}{2m})\sum_{i \in \mathcal{S}(t)} \|\mathbf{w}_i^{(t)} - \mathbf{w}^{(t)}\|^2. \tag{80}$$

We can easily prove that $\frac{v_t(1-\eta_t\mu)}{2\eta_t} \leq 1$, thus we can obtain:

$$
\begin{aligned}
&-\frac{v_t}{m}(1-\eta_t L)\sum_{i\in\mathcal{S}(t)}(F_i(\mathbf{w}^{(t)})-F_i^*)+(\frac{v_t}{2\eta_t m}-\frac{v_t\mu}{2m})\sum_{i\in\mathcal{S}(t)}\|\mathbf{w}_i^{(t)}-\mathbf{w}^{(t)}\|^2 \\
&\leq -\frac{v_t}{m}(1-\eta_t L)\sum_{i\in\mathcal{S}(t)}(F_i(\mathbf{w}^{(t)})-F_i^*)+\frac{1}{m}\sum_{i\in\mathcal{S}(t)}\|\mathbf{w}_i^{(t)}-\mathbf{w}^{(t)}\|^2.
\end{aligned}
\tag{81}
$$

By using Equation 81, we can bound A5 (Equation 75) as:

$$
\begin{aligned}
&\frac{4L\eta_t^2}{m}\mathbb{E}[\sum_{i\in\mathcal{S}(t)}(F_i(\mathbf{w}_i^{(t)})-F_i^*)]-\frac{2\eta_t}{m}\mathbb{E}[\sum_{i\in\mathcal{S}(t)}(F_i(\mathbf{w}_i^{(t)})-F_i(\mathbf{w}^*))] \\
&\leq\frac{1}{m}\mathbb{E}[\sum_{i\in\mathcal{S}(t)}\|\mathbf{w}_i^{(t)}-\mathbf{w}^{(t)}\|^2]-\frac{v_t}{m}(1-\eta_t L)\mathbb{E}[\sum_{i\in\mathcal{S}(t)}(F_i(\mathbf{w}^{(t)})-F_i^*)]+\frac{2\eta_t}{m}\mathbb{E}[\sum_{i\in\mathcal{S}(t)}(F_i(\mathbf{w}^*)-F_i^*)] \\
&\leq 16\eta_t^2\tau^2 G^2-\frac{v_t}{m}(1-\eta_t L)\mathbb{E}[\sum_{i\in\mathcal{S}(t)}(F_i(\mathbf{w}^{(t)})-F_i^*)]+\frac{2\eta_t}{m}\mathbb{E}[\sum_{i\in\mathcal{S}(t)}(F_i(\mathbf{w}^*)-F_i^*)].
\end{aligned}
\tag{82}
$$

Due to the Definition 4.1 and 4.2, we can get:

$$
\begin{aligned}
&16\eta_t^2\tau^2 G^2-\frac{v_t}{m}(1-\eta_t L)\mathbb{E}[\sum_{i\in\mathcal{S}(t)}(F_i(\mathbf{w}^{(t)})-F_i^*)]+\frac{2\eta_t}{m}\mathbb{E}[\sum_{i\in\mathcal{S}(t)}(F_i(\mathbf{w}^*)-F_i^*)] \\
&=16\eta_t^2\tau^2 G^2-v_t(1-\eta_t L)\mathbb{E}[\rho(\mathcal{S}(t),\mathbf{w}^{(t)})(F(\mathbf{w}^{(t)})-\sum_{i=1}^N p_i F_i^*)]+2\eta_t\mathbb{E}[\rho(\mathcal{S}(t),\mathbf{w}^*)(F^*-\sum_{i=1}^N p_i F_i^*)] \\
&\leq 16\eta_t^2\tau^2 G^2-v_t(1-\eta_t L)\overline{\rho}(\mathbb{E}[F(\mathbf{w}^{(t)})]-\sum_{i=1}^N p_i F_i^*)+2\eta_t\widetilde{\rho}\Gamma.
\end{aligned}
\tag{83}
$$

Then, we define:

$$
A_7=-v_t(1-\eta_t L)\overline{\rho}(\mathbb{E}[F(\mathbf{w}^{(t)})]-\sum_{i=1}^N p_i F_i^*).
\tag{84}
$$

We can expand $A_7$ (Equation 84) as:

$$
\begin{aligned}
A_7&=-v_t(1-\eta_t L)\overline{\rho}(\mathbb{E}[F(\mathbf{w}^{(t)})]-\sum_{i=1}^N p_i F_i^*) \\
&=-v_t(1-\eta_t L)\overline{\rho}\sum_{i=1}^N p_i(\mathbb{E}[F_i(\mathbf{w}^{(t)})]-F^*+F^*-F_i^*) \\
&=-v_t(1-\eta_t L)\overline{\rho}\sum_{i=1}^N p_i(\mathbb{E}[F_i(\mathbf{w}^{(t)})]-F^*)-v_t(1-\eta_t L)\overline{\rho}\sum_{i=1}^N p_i(F^*-F_i^*) \\
&=-v_t(1-\eta_t L)\overline{\rho}(\mathbb{E}[F(\mathbf{w}^{(t)})]-F^*)-v_t(1-\eta_t L)\overline{\rho}\Gamma.
\end{aligned}
\tag{85}
$$

Due to the $\mu$-convexity (Assumption 4.2), we can get:

$$
A_7\leq\frac{-v_t(1-\eta_t L)\mu\overline{\rho}}{2}\mathbb{E}[\|\mathbf{w}^{(t)}-\mathbf{w}^*\|^2]-v_t(1-\eta_t L)\overline{\rho}\Gamma.
\tag{86}
$$

We can easily prove that $-2\eta_t(1-2L\eta_t)(1-\eta_t L) \leq -\frac{3}{4}\eta_t$ and $-(1-2L\eta_t)(1-\eta_t L) \leq -(1-3L\eta_t)$, thus we can obtain:

$$A_7 \leq -\frac{3\eta_t\mu\overline{\rho}}{8}\mathbb{E}[\|\mathbf{w}^{(t)} - \mathbf{w}^*\|^2] - 2\eta_t(1-2L\eta_t)(1-\eta_t L)\overline{\rho}\Gamma.$$

$$\leq -\frac{3\eta_t\mu\overline{\rho}}{8}\mathbb{E}[\|\mathbf{w}^{(t)} - \mathbf{w}^*\|^2] - 2\eta_t\overline{\rho}\Gamma + 6\eta_t^2\overline{\rho}L\Gamma. \tag{87}$$

Thus we can bound $A_5$ as:

$$\frac{4L\eta_t^2}{m}\mathbb{E}[\sum_{i\in\mathcal{S}(t)}(F_i(\mathbf{w}_i^{(t)}) - F_i^*) - \frac{2\eta_t}{m}\sum_{i\in\mathcal{S}(t)}(F_i(\mathbf{w}_i^{(t)}) - F_i(\mathbf{w}^*))]$$

$$\leq -\frac{3\eta_t\mu\overline{\rho}}{8}\mathbb{E}[\|\mathbf{w}^{(t)} - \mathbf{w}^*\|^2] + 2\eta_t\Gamma(\widetilde{\rho} - \overline{\rho}) + \eta_t^2(6\overline{\rho}L\Gamma + 16\tau^2 G^2). \tag{88}$$

Now, we can bound $\mathbb{E}[\|\mathbf{w}^{(t+1)} - \mathbf{w}^*\|^2]$ as:

$$\mathbb{E}[\|\mathbf{w}^{(t+1)} - \mathbf{w}^*\|^2] \leq \left[1 - \eta_t\mu(1 + \frac{3\overline{\rho}}{8})\right]\mathbb{E}[\|\mathbf{w}^{(t)} - \mathbf{w}^*\|^2] + \eta_t^2(32\tau^2 G^2 + \frac{\sigma^2}{m} + 6\overline{\rho}L\Gamma) + 2\eta_t\Gamma(\widetilde{\rho} - \overline{\rho}). \tag{89}$$

By defining $\Delta_{t+1} = \mathbb{E}[\|\mathbf{w}^{(t+1)} - \mathbf{w}^*\|^2]$, $B = (1 + \frac{3\overline{\rho}}{8})$, $C = 32\tau^2 G^2 + \frac{\sigma^2}{m} + 6\overline{\rho}L\Gamma$, $D = 2\Gamma(\widetilde{\rho} - \overline{\rho})$, we have that:

$$\Delta_{t+1} \leq (1 - \eta_t\mu B)\Delta_t + \eta_t^2 C + \eta_t D. \tag{90}$$

By setting $\Delta_t \leq \frac{\Theta}{t+\gamma}$, $\eta_t = \frac{\widetilde{\beta}}{t+\gamma}$, $\widetilde{\beta} > \frac{1}{\mu B}$ and $\gamma > 0$, we have that by induction:

$$\Theta = \max\left\{\gamma\|\mathbf{w}^{(0)} - \mathbf{w}^*\|^2, \frac{1}{\widetilde{\beta}\mu B - 1}(\widetilde{\beta}^2 C + D\widetilde{\beta}(t+\gamma))\right\}. \tag{91}$$

Then by the L-smoothness of $F(\cdot)$ (Assumption 4.1), we can get that:

$$\mathbb{E}[F(\mathbf{w}^{(t)})] - F^* \leq \frac{L}{2}\Delta_t \leq \frac{L}{2}\frac{\Theta}{\gamma + t}. \tag{92}$$

For learning rate $\eta = \frac{1}{\mu(t+\gamma)}$ and $\gamma = \frac{4L}{\mu}$, after $T$ iterations, we have the convergence as:

$$\mathbb{E}[F(\mathbf{w}^{(T)})] - F^* \leq \frac{1}{(T+\gamma)}\left[\frac{4L(32\tau^2 G^2 + \sigma^2/m)}{3\mu^2\overline{\rho}} + \frac{8L^2\Gamma}{\mu^2} + \frac{L\gamma\|\mathbf{w}^{(0)} - \mathbf{w}^*\|^2}{2}\right] + \frac{8L\Gamma}{3\mu}(\frac{\widetilde{\rho}}{\overline{\rho}} - 1). \tag{93}$$

## E. Proof of Theorem 4.2

For AFFL, according to the Equation 89 and Equation 90, we can bound $\mathbb{E}[\|\mathbf{w}^{(t+1)} - \mathbf{w}^*\|^2]$ as:

$$\mathbb{E}[\|\mathbf{w}^{(t+1)} - \mathbf{w}^*\|^2] \leq \left[1 - \eta_t\mu(1 + \frac{3\overline{\rho}}{8})\right]\mathbb{E}[\|\mathbf{w}^{(t)} - \mathbf{w}^*\|^2] + \eta_t^2(32\tau^2 G^2 + \frac{\sigma^2}{m} + 6\overline{\rho}L\Gamma) + 2\eta_t\Gamma(\widetilde{\rho} - \overline{\rho}). \tag{94}$$

$$\Delta_{t+1} \leq (1 - \eta_t\mu B)\Delta_t + \eta_t^2 C + \eta_t D. \tag{95}$$

$$B = 1 + \frac{3\overline{\rho}}{8}, C = 32\tau^2 G^2 + \frac{\sigma^2}{m} + 6\overline{\rho}L\Gamma, D = 2\Gamma(\widetilde{\rho} - \overline{\rho}). \tag{96}$$

Because AFFL lacks local multi-step updates, so, we can get $\tau = 0$, and:

$$C = \frac{\sigma^2}{m} + 6\overline{\rho}L\Gamma. \tag{97}$$

Since all clients participate in training, $\widetilde{\rho} = \overline{\rho} = 1$. $\widetilde{\rho} = \overline{\rho} = 1$, then we can get:

$$B = \frac{11}{8}, D = 0. \tag{98}$$

The convergence behavior of centralized training on the real dataset is given by:

$$\mathbb{E}[\|\mathbf{w}^{(t+1)} - \mathbf{w}^*\|^2] \leq \left[1 - \frac{11\eta_t\mu}{8}\right] \mathbb{E}[\|\mathbf{w}^{(t)} - \mathbf{w}^*\|^2] + \eta_t^2(\frac{\sigma^2}{m} + 6L\Gamma). \tag{99}$$

By decomposing the stochastic gradient $\mathbf{g}(\mathbf{w}^{(t)}, \widetilde{\xi}^{(t)})$ ($\widetilde{\xi}^{(t)}$ is mini-batch uniformly sampled at random from $\widetilde{D}^t$), we obtain:

$$\mathbf{g}(\mathbf{w}^{(t)}, \widetilde{\xi}^{(t)}) = \nabla F(\mathbf{w}^{(t)}) + (\nabla \widetilde{F}(\mathbf{w}^{(t)}, \widetilde{D}^t) - \nabla F(\mathbf{w}^{(t)})) + (\mathbf{g}(\mathbf{w}^{(t)}, \widetilde{\xi}^{(t)}) - \nabla \widetilde{F}(\mathbf{w}^{(t)}, \widetilde{D}^t)), \tag{100}$$

where $\nabla F(\mathbf{w}^{(t)})$ represent gradient generated by real data, $(\nabla \widetilde{F}(\mathbf{w}^{(t)}, \widetilde{D}^t) - \nabla F(\mathbf{w}^{(t)}))$ denotes the gradient bias of the synthetic data, $(\mathbf{g}(\mathbf{w}^{(t)}, \widetilde{\xi}^{(t)}) - \nabla \widetilde{F}(\mathbf{w}^{(t)}, \widetilde{D}^t))$ denotes the bias introduced by training sampling.

In FL, the bias term is only from the sampling noise. However, under the AFFL setting, it is necessary to additionally incorporate the bias caused by data distillation. To this end, we begin by introducing the following assumption:

**Assumption E.1 (Bounded Synthetic Bias).** The bias introduced by data distillation admits a minimal upper bound:

$$\|\nabla \widetilde{F}(\mathbf{w}^{(t)}, \widetilde{D}^t) - \nabla F(\mathbf{w}^{(t)})\|^2 \leq \zeta_t^2. \tag{101}$$

According to the Assumption E.1, we need to replace the client training sampling bias $\sigma^2$ in Equation 99 with the sum of server sampling bias and synthesis bias $\widetilde{\sigma}^2 + \zeta_t^2$. Since the server are relatively powerful, they are often able to use larger batch sizes for training, so $\widetilde{\sigma}^2 \leq \frac{\sigma^2}{m}$, and we get a new upper bound for AFFL:

$$\mathbb{E}[\|\mathbf{w}^{(t+1)} - \mathbf{w}^*\|^2] \leq \left[1 - \frac{11\eta_t\mu}{8}\right] \mathbb{E}[\|\mathbf{w}^{(t)} - \mathbf{w}^*\|^2] + \eta_t^2(\widetilde{\sigma}^2 + \zeta_t^2 + 6L\Gamma), \tag{102}$$

$$\Delta_{t+1} \leq (1 - \eta_t\mu\frac{11}{8})\Delta_t + \eta_t^2(\widetilde{\sigma}^2 + \zeta_t^2 + 6L\Gamma), \eta_t < \frac{1}{4L}. \tag{103}$$

By setting $\Delta_t \leq \frac{\Theta}{t+\gamma}, \eta_t = \frac{\widetilde{\beta}}{t+\gamma}, \widetilde{\beta} > \frac{8}{11\mu}$ and $\gamma > 0$, we have that by introduction:

$$\Theta = \max\{\gamma\|\mathbf{w}^{(0)} - \mathbf{w}^{(t)}\|^2, \frac{\widetilde{\beta}^2(\widetilde{\sigma}^2 + 6L\Gamma + \zeta_t^2)}{\frac{11}{8}\mu\widetilde{\beta} - 1}\}. \tag{104}$$

Then by the L-smoothness of F $(\cdot)$ (Assumption 4.1), we can get that:

$$\mathbb{E}[F(\mathbf{w}^{(t)})] - F^* \leq \frac{L}{2}\Delta_t \leq \frac{L}{2}\frac{\psi}{\gamma + t}. \tag{105}$$

For learning rate $\eta = \frac{1}{\mu(t+\gamma)}$ and $\gamma = \frac{4L}{\mu}$, after $T$ iterations, we have the convergence as:

$$\mathbb{E}[F(\mathbf{w}^{(T)})] - F^* \leq \frac{1}{(T+\gamma)}\left[\frac{32L}{33\mu^2}(\widetilde{\sigma}^2 + \zeta_T^2) + \frac{8L^2\Gamma}{\mu^2} + \frac{L\gamma\|\mathbf{w}^{(0)} - \mathbf{w}^*\|^2}{2}\right]. \tag{106}$$

# F. Convergence Comparison Between FL and AFFL

In summary, when learning rate $\eta = \frac{1}{\mu(t+\gamma)}$ and $\gamma = \frac{4L}{\mu}$, after $T$ iterations, the convergence result of conventional FL is given by:

$$\mathbb{E}[F(\mathbf{w}^{(T)})] - F^* \leq \frac{1}{(T+\gamma)}\left[\frac{4L(32\tau^2G^2 + \sigma^2/m)}{3\mu^2\overline{\rho}} + \frac{8L^2\Gamma}{\mu^2} + \frac{L\gamma\|\mathbf{w}^{(0)} - \mathbf{w}^*\|^2}{2}\right] + \frac{8L\Gamma}{3\mu}(\frac{\widetilde{\rho}}{\overline{\rho}} - 1), \tag{107}$$

and the convergence result of conventional AFFL is given by:

$$\mathbb{E}[F(\mathbf{w}^{(T)})] - F^* \leq \frac{1}{(T+\gamma)} \left[ \frac{32L}{33\mu^2}(\widetilde{\sigma}^2 + \zeta_T^2) + \frac{8L^2\Gamma}{\mu^2} + \frac{L\gamma\|\mathbf{w}^{(0)} - \mathbf{w}^*\|^2}{2} \right]. \tag{108}$$

Compared with FL, AFFL eliminates the constant bias term that does not decay with $T$, so it leads to stronger convergence. In the following, we establish the necessary and sufficient conditions for AFFL to outperform traditional FL in terms of convergence.

$$\widetilde{\sigma}^2 + \zeta_T^2 \leq \frac{11}{8\overline{\rho}}(\frac{\sigma^2}{m} + 32\tau^2G^2), \overline{\rho} < 1. \tag{109}$$

The error of synthetic data originates from both sampling bias $\widetilde{\sigma}^2$ and synthesis bias $\zeta_t^2$. Both of these can be substantially reduced in practice. Thanks to the strong computational capacity of server, we can employ a large batch size on the server side to significantly reduce the sampling bias $\widetilde{\sigma}^2$. Moreover, in Section 3.2, we have already established Theorem 3.1, which demonstrates the unbiased of our synthetic data. Therefore, the overall error of FedUSD approaches zero. In contrast, the error in FL is often uncontrollable. The limited computational capacity of clients typically leads to large sampling bias $\frac{\sigma^2}{m}$. Moreover, due to the low communication efficiency, FL usually adopts a larger number of local update steps to reduce communication frequency, which in turn introduces substantial multi-step drift $32\tau^2G^2$.

Next, we compute the crossover round. We denote Equation 107 and Equation 108 respectively as:

$$\text{FL} = \frac{A_{\text{FL}}}{T+\gamma} + \mathcal{B}, \text{AFFL} = \frac{A_{\text{AFFL}}}{T+\gamma}, \mathcal{B} = \frac{8L\Gamma}{3\mu}(\frac{\widetilde{\rho}}{\rho} - 1). \tag{110}$$

If $A_{\text{AFFL}} > A_{\text{FL}}$, AFFL will still outperform after sufficient rounds, and the required number of rounds is:

$$T^* = [\frac{A_{\text{AFFL}} - A_{\text{FL}}}{\mathcal{B}} - \gamma], A_{\text{AFFL}} = \frac{32L}{33\mu^2}(\widetilde{\sigma}^2 + \zeta_T^2), A_{\text{FL}} = \frac{4L(32\tau^2G^2 + \sigma^2/m)}{3\mu^2\overline{\rho}}. \tag{111}$$

If $A_{\text{AFFL}} < A_{\text{FL}}$, AFFL is superior starting from the very first round.

## G. Differential Privacy of FedUSD

Privacy preservation is an essential factor when evaluating methods in FL. Inspired by FedDM (Xiong et al., 2023), we conduct a theoretical analysis of the privacy guarantees of FedUSD. Before analyzing FedUSD, we first present the fundamental principles of differential privacy.

**Definition G.1 (Differential Privacy** (Dwork et al., 2006)**).** A randomized mechanism M: $\mathcal{D} \to \mathcal{R}$ with domain $\mathcal{D}$ and range $\mathcal{R}$ satisfies $(\epsilon, \delta)$-differential privacy if for any two adjacent dataset $D_1, D_2$ and any measurable subset $S \subseteq \mathcal{R}$,

$$\mathbf{Pr}(\text{M}(D_1) \in S) \leq e^\epsilon \mathbf{Pr}(\text{M}(D_2) \in S) + \delta. \tag{112}$$

In this paper, we focus on instance-level differential privacy, where a randomized mechanism is applied to the query function of a dataset $f : \mathcal{D} \to \mathcal{X}$. Without loss of generality, we consider output spaces $\mathcal{R}, \mathcal{X} \subseteq \mathbb{R}^m$. The central concept in the analysis of differential privacy is the query sensitivity, which measures the worst-case deviation in the query output resulting from a one-instance variation in the dataset. For a query function $f : \mathcal{D} \to \mathbb{R}^m$ and $\ell_p$ norm the sensitivity is formally defined as:

$$\Delta_p \triangleq \max_{D_1, D_2} \|f(D_1) - f(D_2)\|_p. \tag{113}$$

We employ the Gaussian mechanism to achieve differential privacy,

$$\text{M}(D) \triangleq f(D) + Z, \text{where } Z \sim \mathcal{N}(0, \sigma_{\text{DP}}^2\Delta_p^2\mathbf{I}), \tag{114}$$

where the differential privacy is satisfied for the function $f$ of sensitivity $\Delta_p$ if we choose $\sigma_{\text{DP}} \geq \sqrt{2\frac{\log(1.25/\delta)}{\epsilon}}$. Differentially Private SGD (DP-SGD) applies the Gaussian mechanism throughout hundreds of optimization steps, and we can get:

**Theorem G.1 (Differential Privacy of DP-SGD (Xiong et al., 2023).).** *There exist constants $c_1$ and $c_2$ so that given the sampling probability $q$ and the number of steps $E$, for any $\epsilon < c_1 q^2 E$, DP-SGD is $(\epsilon, \delta)$-differential privacy for any $\delta > 0$ if*

$$\sigma \geq c_2 \frac{q\sqrt{E \log(1/\delta)}}{\epsilon}. \tag{115}$$

Then, we prove that by updating the synthetic dataset using DP-SGD, FedUSD can guarantee differential privacy. We begin by analyzing each client separately in order to assess its differential privacy guarantees, we prove that the gradient of $\mathcal{L}_{USD}$ can be expressed as the average of the individual gradients of each real example.

$$\mathcal{L}_{USD} = \lambda_1 \mathcal{L}_{\text{HOB}} + \lambda_2 \mathcal{L}_{\text{Var}} = \lambda_1 \text{EMD}(\mathbf{v}_{ik}, \widetilde{\mathbf{v}}_{ik}^t) + \lambda_2 \text{EMD}(\psi_{ik}, \widetilde{\psi}_{ik}^t). \tag{116}$$

Let $\pi^*$ serve as the optimal transport plan, for each column (the $n$-th "anchor") of $\widetilde{\mathbf{v}}_{ik}^t$ or $\widetilde{\psi}_{ik}^t$, we have:

$$[\nabla_{\widetilde{a}} \text{EMD}(a, \widetilde{a})]_n = \sum_{m=1}^{M} \pi^*_{mn} \frac{\widetilde{a}_n - a_m}{\|\widetilde{a}_n - a_m\|_2 + \varpi}, \ a \in \{\mathbf{v}_{ik}, \psi_{ik}\}, \ \widetilde{a} \in \{\widetilde{\mathbf{v}}_{ik}^t, \widetilde{\psi}_{ik}^t\}. \tag{117}$$

We first rewrite the variance path $\mathcal{L}_{Var}$ strictly as a per-real-sample average:

$$\psi_{ik} = \frac{1}{|\xi_{ik}|} \sum_{x_{ij,k} \in \xi_{ik}} \text{Var}(x_{ij,k}), \ \text{Var}(x_{ij,k}) = (z_{ij,k} - \mu_{ik})(z_{ij,k} - \mu_{ik})^\top, \tag{118}$$

Where $\xi_{ik}$ is the real samples selected by sampling for class $k$ on client $i$. According to the chain rule, we obtain that:

$$\begin{aligned}
\nabla_{\widetilde{D}_{ik}} \mathcal{L}_{Var} &= J_{\widetilde{D}_{ik}}(\widetilde{\psi}_{ik}) \nabla_{\widetilde{\psi}_{ik}} \text{EMD}(\psi_{ik}, \widetilde{\psi}_{ik}) \\
&= J_{\widetilde{D}_{ik}}(\widetilde{\psi}_{ik}) \sum_{n=1}^{\mathcal{M}} \Big( \sum_{m=1}^{\mathcal{M}} \pi^*_{mn} \frac{\widetilde{\psi}_{ik,n} - \frac{1}{|\xi_{ik}|} \sum_{x_{ij,k} \in \xi_{ik}} \text{Var}(x_{ij,k})_m}{\|\widetilde{\psi}_{ik,n} - \psi_{ik,m}\|_2 + \varpi} \Big) e_n \\
&= \frac{1}{|\xi_{ik}|} \sum_{x_{ij,k} \in \xi_{ik}} J_{\widetilde{D}_{ik}} \sum_{n=1}^{\mathcal{M}} \Big( \sum_{m=1}^{\mathcal{M}} \pi^*_{mn} \frac{\widetilde{\psi}_{ik,n} - \text{Var}(x_{ij,k})_m}{\|\widetilde{\psi}_{ik,n} - \psi_{ik,m}\|_2 + \varpi} \Big) e_n \\
&= \frac{1}{|\xi_{ik}|} \sum_{x_{ij,k} \in \xi_{ik}} g_{Var}(\mathbf{x}_{ij,k}), \ g_{Var}(\mathbf{x}_{ij,k}) = J_{\widetilde{D}_{ik}} \sum_{n=1}^{\mathcal{M}} \Big( \sum_{m=1}^{\mathcal{M}} \pi^*_{mn} \frac{\widetilde{\psi}_{ik,n} - \text{Var}(x_{ij,k})_m}{\|\widetilde{\psi}_{ik,n} - \psi_{ik,m}\|_2 + \varpi} \Big) e_n,
\end{aligned} \tag{119}$$

where $J_{\widetilde{D}_{ik}}$ denotes the Jacobian operator, and it corresponds to the per-real-sample average: $\nabla_{\widetilde{D}_{ik}} \mathcal{L}_{Var} = \frac{1}{|\xi_{ik}|} \sum_{\mathbf{x}_{ij,k} \in \xi_{ik}} g_{Var}(\mathbf{x})$.

Next, we rewrite the HOB path $\mathcal{L}_{HOB}$ strictly as a per-real-sample average. For it, we first assume that the covariance of real data is given by $\mathbf{Cov} = \frac{1}{|\xi_{ik}|} \sum_{i=1}^{|\xi_{ik}|} (\mathbf{z}_{ik} - \mu_{ik})(\mathbf{z}_{ik} - \mu_{ik})^\top$, it can be decomposed into its eigenvalues and eigenvectors as follows: $\mathbf{Cov} = \sum_{j=1}^{\mathcal{M}} \lambda_{Cov}^j \mathbf{v}_{ik}^j \mathbf{v}_{ik}^{j\top}, \mathbf{v}_{ik}^j \in \mathbf{V}_{ik}^\top$. By applying the first-order Fréchet derivative with respect to spectral perturbation, when $\mathbf{Cov} \to \mathbf{Cov} + \Delta\mathbf{Cov}$, we have:

$$\mathbb{D}_{\mathbf{v}_{ik}}[\Delta\mathbf{Cov}] = \sum_{j \neq 1} \frac{\mathbf{v}_{ik}^j \mathbf{v}_{ik}^{j\top} (\Delta\mathbf{Cov}) \mathbf{v}_{ik}}{\lambda_{Cov}^1 - \lambda_{Cov}^j}. \tag{120}$$

We define the contribution of a single real sample as: $\Delta\mathbf{Cov}(\mathbf{x}_{ij,k}) := \text{Var}(\mathbf{x}_{ij,k}) - \psi_{ik}$, then we have: $\phi_{HOB}(\mathbf{x}_{ij,k}) := \mathbb{D}_{\mathbf{v}_{ik}}[\Delta\mathbf{Cov}(\mathbf{x}_{ij,k})]$, and $\frac{1}{|\xi_{ik}|} \sum_{\mathbf{x}_{ij,k} \in \xi_{ik}} \phi_{HOB}(\mathbf{x}_{ij,k}) \approx 0$. Thus the first-order approximation of $\mathbf{v}_{ik}$ with respect to $\mathbf{Cov}$ is given by: $\mathbf{v}_{ik} \approx \overline{\mathbf{v}}_{ik} + \frac{1}{|\xi_{ik}|} \sum_{\mathbf{x}_{ij,k} \in \xi_{ik}} \phi_{HOB}(\mathbf{x}_{ij})$.

Because the $\lambda_{Cov}^1$ is well separated from the others, and the approximation remains sufficiently stable, using $\phi_{HOB}(\mathbf{x}_{ij,k})$ allows the principal direction to be distributed across individual samples. Then for $\mathcal{L}_{HOB}$, we obtain by the chain rule:

$$\nabla_{\widetilde{D}_{ik}} \mathcal{L}_{HOB} = J_{\widetilde{D}_{ik}}(\widetilde{\mathbf{v}}_{ik}) \nabla_{\widetilde{\mathbf{v}}_{ik}} \text{EMD}(\mathbf{v}_{ik}, \widetilde{\mathbf{v}}_{ik}). \tag{121}$$

We set $\Delta \mathbf{Cov}(\mathbf{x}_{ij,k}) := \mathrm{Var}(\mathbf{x}_{ij,k}) - \psi_{ik}$, and Substitute $\mathbf{v}_{ik} \approx \overline{\mathbf{v}}_{ik} + \frac{1}{|\xi_{ik}|} \sum_{\mathbf{x}_{ij,k} \in \xi_{ik}} \phi_{HOB}(\mathbf{x}_{ij})$ into Equation 121, we obtain:

$$
\begin{aligned}
\nabla_{\widetilde{D}_{ik}} \mathcal{L}_{HOB} &= J_{\widetilde{D}_{ik}}(\widetilde{\mathbf{v}}_{ik}) \sum_{n=1}^{\mathcal{M}} \left( \sum_{m=1}^{\mathcal{M}} \pi_{mn}^* \frac{\widetilde{\mathbf{v}}_{ik}^n - \overline{\mathbf{v}}_{ik}^m - \frac{1}{|\xi_{ik}|} \sum_{\mathbf{x}_{ij,k} \in \xi_{ik}} \phi_{HOB}(\mathbf{x}_{ij,k})}{\|\widetilde{\mathbf{v}}_{ij,k}^n - \mathbf{v}_{ij,k}^m\|_2 + \varpi} \right) e_n + \Delta_{\mathrm{2nd}} \\
&= \frac{1}{|\xi_{ik}|} \sum_{x_{ij,k} \in \xi_{ik}} J_{\widetilde{D}_{ik}}(\widetilde{\mathbf{v}}_{ik}) \sum_{n=1}^{\mathcal{M}} \left( \sum_{m=1}^{\mathcal{M}} \pi_{mn}^* \frac{\widetilde{\mathbf{v}}_{ik}^n - \overline{\mathbf{v}}_{ik}^m - \phi_{HOB}(\mathbf{x}_{ij,k})}{\|\widetilde{\mathbf{v}}_{ij,k}^n - \mathbf{v}_{ij,k}^m\|_2 + \varpi} \right) e_n + \Delta_{\mathrm{2nd}} \\
&= \frac{1}{|\xi_{ik}|} \sum_{x_{ij,k} \in \xi_{ik}} g_{HOB}(\mathbf{x}_{ij,k}) + \Delta_{\mathrm{2nd}}, \\
g_{HOB}(\mathbf{x}_{ij,k}) &= J_{\widetilde{D}_{ik}}(\widetilde{\mathbf{v}}_{ik}) \sum_{n=1}^{\mathcal{M}} \left( \sum_{m=1}^{\mathcal{M}} \pi_{mn}^* \frac{\widetilde{\mathbf{v}}_{ik}^n - \overline{\mathbf{v}}_{ik}^m - \phi_{HOB}(\mathbf{x}_{ij,k})}{\|\widetilde{\mathbf{v}}_{ij,k}^n - \mathbf{v}_{ij,k}^m\|_2 + \varpi} \right) e_n,
\end{aligned}
\tag{122}
$$

where, the term $\Delta_{\mathrm{2nd}}$ denotes the second-order remainder arising from the first-order approximation of $\mathbf{v}_{ik}^m$, Since the principal energy is highly concentrated and the spectral gap is sufficiently large, $\Delta_{\mathrm{2nd}}$ can be regarded as a small term, which does not affect the Differential Privacy accounting.

Therefore, by combining the two paths, we obtain the per-real-sample average of the total USD gradient $g_{USD}$.

$$
g_{USD}(\mathbf{x}_{ij,k}) = \lambda_1 g_{HOB} + \lambda_2 g_{Var},
\tag{123}
$$

$$
\nabla_{\widetilde{D}_{ik}} \mathcal{L}_{USD} = \frac{1}{|\xi_{ik}|} \sum_{\mathbf{x}_{ij,k} \in \xi_{ik}} g_{USD}(\mathbf{x}_{ij,k}) + \lambda_1 \Delta_{\mathrm{2nd}} \approx \frac{1}{|\xi_{ik}|} \sum_{\mathbf{x}_{ij,k} \in \xi_{ik}} g_{USD}(\mathbf{x}_{ij,k}).
\tag{124}
$$

It indicates that the synthetic data can be regarded as the equivalent of the network parameters in DP-SGD. Consequently, we conclude that for each client $i$, Theorem G.1 holds when optimizing $\widetilde{D}_{ik}$. Next, in order to extend the DP guarantee to a system with $N$ clients, we apply the parallel composition (McSherry, 2009):

**Theorem G.2 (Parallel Composition (McSherry, 2009).).** *Suppose there are $N$ mechanisms $\mathrm{M}_1, ..., \mathrm{M}_N$ operating on mutually disjoint subsets, each providing $(\epsilon_i, \delta_i)$-DP. Then any combined function of $\mathrm{M}_1, ..., \mathrm{M}_N$ preserves differential privacy with parameters $(\max_i \epsilon_i, \max_i \delta_i)$.*

We observe that each client maintains its own local dataset, satisfying the property of mutual disjoint-ness. Under this setting, if every client ensures $(\epsilon, \delta)$-differential privacy individually, then the Gaussian mechanism remains $(\epsilon, \delta)$-differential privacy for the entire system by the parallel composition principle. Furthermore, to quantify the amount of noise required for each client, we can employ the Tail bound (Abadi et al., 2016):

$$
\delta = \min_{\Lambda} \exp\left( \alpha_M(\Lambda) - \Lambda \epsilon \right),
\tag{125}
$$

where $\alpha_M(\Lambda) \leq Eq^2\Lambda^2/\sigma_{DP}^2$, without loss of generality, set $\Lambda = \sigma_{DP}^2$, it holds that $\delta \leq \exp\left( Eq^2\sigma_{DP}^2 - \epsilon\sigma_{DP}^2 \right)$, and $\sigma_{DP} \geq \sqrt{\frac{\log(1/\delta)}{Eq^2 - \epsilon}}$. When $Eq^2 \leq \frac{\epsilon}{2}$, $\sigma_{DP} \geq \sqrt{\frac{2\log(1/\delta)}{\epsilon}}$.

To ensure differential privacy when using the synthetic dataset for downstream tasks, we introduce the following post-processing property (Dwork et al., 2014):

**Theorem G.3 (Robustness to Post-Processing (Dwork et al., 2014).).** *Suppose $\mathrm{M}$ is a randomized mechanism that satisfies $(\epsilon, \delta)$-differential privacy. Then, for any (possibly randomized) function $F_{DP}$, the transformed output $F_{DP}(\mathrm{M})$ preserves the same $(\epsilon, \delta)$-differential privacy guarantee.*

Training a network on the synthetic dataset $\widetilde{D}_{ik}$ is a post-processing operation. According to Theorem G.3, as long as the generation of $\widetilde{D}_{ik}$ satisfies $(\epsilon, \delta)$-differential privacy, any subsequent computation or post-processing performed on $\widetilde{D}_{ik}$ also remains differentially private. Finally, with Theorem G.1, G.2 and G.3, we complete the proof of Theorem G.4.

**Theorem G.4 (Differential Privacy Guarantee of FedUSD.).** *Given that the synthetic dataset $\widetilde{D}_{ik}$, FedUSD trained with DP-SGD ensures $(\epsilon, \delta)$-differential privacy in FL with $N$ clients. The privacy guarantee holds if the noise multiplier $\sigma_{DP}$ in*

*each round satisfies:*

$$\sigma_{DP} \geq \sqrt{\frac{\log(\delta)}{Eq^2 - \epsilon}}, \text{ or } \sigma_{DP} \geq \sqrt{\frac{2\log(1/\delta)}{\epsilon}} \text{ when } Eq^2 \leq \epsilon/2. \tag{126}$$

For the total DP budget over $R$ communication rounds, assume that in each round the Gaussian mechanism provides $(\epsilon, \delta)$-differetial privacy. According to the advanced composition theorem (Kairouz et al., 2015), performing such queries results in an overall privacy guarantee of

$$(O(\sqrt{R\log(1/\delta')})\epsilon, R\delta + \delta') - \text{Differential Privacy}, \forall \delta' \in (0, 1/2), \tag{127}$$

which implies a moderate accumulation of the privacy budget across rounds.

