# OpenReview forum: "FedUSD: Unbiased Synthetic Data for Federated Learning"
_ICML.cc/2026/Conference — ICML 2026 regular_

### Official Review · Reviewer_hoBj · 2026-03-12

**Soundness:** 3
**Presentation:** 3
**Significance:** 3
**Originality:** 3
**Overall Recommendation:** 4
**Confidence:** 3

**Summary:**

The paper proposes Federated Unbiased Synthetic Data (FedUSD), an aggregation-free FL algorithm that avoids the bias introduced by the residual components. By utilizing the observation that the hidden feature's energy is concentrated in the first orthogonal direction, this paper proposes a novel loss function involving HOB loss. They prove that the HOB reconstruction error is smaller than or equal to the mean-based reconstruction error, which is a cornerstone of the aforementioned loss function and motivates the development of the FedUSD algorithm. They provide both theoretical and empirical studies of FedUSD, showing an advantage compared to other methods.

**Compliance With Llm Reviewing Policy:**

Affirmed.

**Final Justification:**

Addressed my concerns, so I keep the positive score.

**Key Questions For Authors:**

See W3, what is the advantage of the convergence rate of AFFL with respect to $T$? And, what role does $\mathcal{B}$ play here? I will increase from 4 to 5 if these are addressed.

**Limitations:**

yes

**Strengths And Weaknesses:**

**Strengths**:

1. The idea of this paper is very clear and straightforward to perceive. The structure starts from motivation, then methodology, and theoretical guarantees and empirical studies.

2. The empirical studies are comprehensive and convincing.

**Weaknesses**:

1. There are some definition/presentation issues in the paper, here I list some of them:

(a) It is better to specify the dimensions when introducing variables, such as, $\mathbf{Z}, \mathbf{z}, \beta_{ik}, g_{ik}$.

(b) Please specify equation (6) is singular value decomposition.

(c) Please specify what are the $\lambda$'s in equation (5).


2. The USD loss is constructed by a linear combination of $L_{HOB}$ and $L_{Var}$.
$L_{HOB}$ is clear from the context, however, the motivation of involving $L_{Var}$ is not presented.

3. Theorems 4.7 and 4.8. The learning rate $\eta$ depends on $t$, it is recommended to use $\eta\_t = \frac{1}{\mu (t+\gamma)}$. The definition of the parameter $\tau$ is missing. The authors would want to elaborate on the setups of FL and AFFL. Notably, Theorem 4.7 is **identical** to Theorem 3.1 in (Cho et al. 2022). The statement of Theorem 4.8 is incomplete either, as it requires an additional Assumption E.1 (lines 1118 - 1120).

4. RHS of line 57 and LHS of line 257 mention that "FedUSD satisfies differential privacy." However, the analysis in Appendix G is limited to the amount of noise required to make FedUSD private. The impacts of noise on the convergence remain unclear. Furthermore, no numerical experiments are conducted to evaluate the privacy.

4. For equation (24), authors mention that $\zeta_T^2$ is the upper bound of bias introduced by data distillation, however, since it depends on $T$, I would like to see its definition or order w.r.t. $T$ in the theorem statement. Otherwise it is not clear what is the order of the AFFL convergence upper bound w.r.t. $T$ . Besides, it seems AFFL has advantage over FL up to a constant ($A_{\text{AFFL}}$ vs $A_{\text{FL}}$) if ignoring $\mathcal{B}$, so could the author give an explanation of how significant the advantage is?

Minors:
1. what is "produce description" in the title of Section 3.2?

---

> ### Author Rebuttal · Authors · 2026-03-31
>
> Dear reviewer, thank you for recognizing the contributions of our paper and your valuable feedback. We sincerely hope our responses address your concerns and we look forward to engaging in constructive discussion with you.
>
> 1.	Definition and Presentation issues.
> We agree that several definitions and notations should be clarified. In the revision, we will explicitly add missing variable dimensions, clarify that Eq. (6) is the SVD of the hidden representation set; specify the $\lambda_1, \lambda_2$ are weighting coefficients; and fix presentation issues such as the typo in Section 3.2 (“Produce Description” to “Process Description”).
> 2.	Motivation for $L_{Var}$.
> Thank you for your suggestions. We agree that the motivation of $L_{Var}$ should be clarified clearer. FedUSD mainly targets the principal component, because existing mean-based matching introduces the most direct bias. Therefore, the main novelty of FedUSD lies in replacing mean matching with HOB matching. By contrast, for the residual component, our goal is not to fully characterize all residual structure, but to provide a lightweight and stable constraint that prevents the synthetic features from collapsing excessively toward the class center. We will revise the paper to make the motivation explicit.
> 3.	Theoretical notation and assumptions.
> Thank you for your suggestions. We agree that several notational issues should be cleaned up. In particular, we will make the learning-rate notation more precise, we will also clarify that $\tau$ denotes the synchronization period in the FL, and ensure that all assumptions required by Theorem 4.8 are clearly stated in the theorem statement. Regarding Theorem 4.7, our intention is to use the standard FL side result as a reference point for a unified comparison, while the AFFL side result and the crossover round analysis are the main parts specific to our paper. We will make the connection with prior work clearer in the revised version.
> 4.	DP-SGD convergence and privacy experiment.
> Thank you for your suggestions. We agree that the effect of DP noise on convergence should be made explicit. Building on Theorem 4.8, it can be captured by introducing an additional term $\Xi_{DP}^2$ for the extra distillation error caused by DP-SGD: $\mathbb{E}[F(w^{(T)})]-F^{\*} \\leq \frac{1}{(T+\gamma)} [\frac{32L}{33\mu^2}(\widetilde{\sigma}^2+\zeta_T^2+\Xi_{DP}^2)+ \frac{8L^2\Gamma}{\mu^2}+\frac{L\gamma||w^{(0)}-w^*||^2}{2}].$ Using the local synthetic optimization round $E$ as the DP-SGD step number, we can upper bound: $\Xi_{DP}^2 \leq dim(f)\Delta_p^2 c_2^2\frac{q^2 E \log(1/\delta)}{\epsilon^2}$, $dim(f)$ is the dimension of noise. Details will be in the revision. In addition, we further evaluate the privacy-utility trade-off of FedUSD on CIFAR-10 with ConvNet under different privacy budgets when $\alpha=0.1$. As shown in table, FedUSD still maintains strong utility under moderate privacy budgets.
> | Methods | σ_DP=1 | σ_DP=3 | σ_DP=5 |
> |---|---:|---:|---:|
> | FedDM | 63.42 | 61.51 | 58.17 |
> | FedAF | 66.92 | 65.64 | 63.21 |
> | **FedUSD** | **73.96** | **72.42** | **70.38** |
>
> 5.	Meaning of $\zeta_t^2$, dependence on T and role of $\mathcal{B}$.
> Thank you for your suggestions. First, $\zeta_t^2$ is defined in Ass. E.1 as the upper bound on the gradient bias introduced by synthetic data optimization. Therefore, in Theorem 4.8 / Eq. (24), $\zeta_T^2$ should be interpreted as the worst-case synthetic bias up to round $T$ : $\zeta_T^2=\max_{1\leq t\leq T}\zeta_t^2$ . We agree that this is not stated clearly enough, and we will revise Eq. (24) and the surrounding text to make it explicit.
> Second, under the additional bounded-bias condition that $\zeta_t^2$ is uniformly bounded over $t$, the order of $\zeta_T^2$ with respect to $T$ is $\mathcal{O}(1)$. It therefore only affects the numerator constant in the AFFL bound, while the convergence order in $T$ remains exactly $\mathcal{O}(\frac{1}{T+\gamma})$.
> Third, $\mathcal{B}$ is the FL-only non-decaying error floor caused by data heterogeneity and client-selection bias under partial participation. By contrast, AFFL is naturally more compatible with full client participation, since smaller communication costs and the main global model training is performed centrally on the server. As a result, AFFL avoids the client-selection bias introduced by partial participation and removes the corresponding term $\mathcal{B}$ from the convergence bound. Even if $\mathcal{B}$ is temporarily ignored, there is still practical reason to expect $ A_{AFFL}<A_{FL}$ : the server can use stronger computation and larger batch sizes to reduce $\widetilde{\sigma}^2$, while FedUSD is designed to reduce $\zeta_T^2$; in contrast, FL typically suffers from larger client-side variance $\sigma^2/m$ and the local-update drift term $32 \tau^2G^2$. Therefore, AFFL benefits both from a smaller transient term in practice and from removing the non-decaying term $\mathcal{B}$.

---

> > ### Author Rebuttal · Reviewer_hoBj · 2026-04-02
> >
> > A follow-up question for W4 is, Theorem G.2 points out $\sigma$ depends on $T$, then in the response why does $\Xi_{DP}^2$ not depend on $T$?

---

> > > ### Author Response · Authors · 2026-04-02
> > >
> > > Dear reviewer, thank you for your valuable feedback, and we apologize that this distinction was not stated clearly enough in our previous rebuttal. In Theorem G.2, the symbol $T$ denotes the number of DP-SGD steps in a single noisy synthetic-data optimization, which corresponds to the local synthetic-data optimization round $E$ in our setting. This is because, at each communication round, clients re-sample the data and start a new noisy optimization process for the synthetic set. By contrast, in the convergence analysis (Theorems 4.7 and 4.8), $T$ denotes the number of communication rounds. Therefore, the privacy accountant in Theorem G.2 applies to the within-round DP-SGD procedure, and its step count should be governed by $E$, rather than the outer communication-round variable $T$. Our previous rebuttal and the current manuscript did not distinguish these two meanings clearly enough. Accordingly, the intended DP-error term should depend on $E$, not on the communication-round $T$, i.e., $\Xi_{DP}^2 \leq dim(f)\Delta_p^2 c_2^2\frac{q^2 E \log(1/\delta)}{\epsilon^2}$. We will correct this notation explicitly in the revised version to avoid ambiguity and make the symbol usage consistent.

---

### Official Review · Reviewer_45bK · 2026-03-12

**Soundness:** 3
**Presentation:** 2
**Significance:** 2
**Originality:** 3
**Overall Recommendation:** 4
**Confidence:** 4

**Summary:**

This paper studies aggregation-free federated learning via synthetic data sharing and proposes a synthetic data optimization method FedUSD.  The main idea is to separately capture the principal and residual components of client data in feature space through high-energy orthogonal basis and variance matching. Experiments on CIFAR-10, CIFAR-100, and SVHN report consistent gains over some baselines.

**Compliance With Llm Reviewing Policy:**

Affirmed.

**Final Justification:**

The rebuttal provides additional clarification on the main points raised in my review. While some aspects could still benefit from further refinement, the response helps clarify the authors’ position and addresses the key issues to a reasonable extent. I have updated the corresponding assessment accordingly.

**Key Questions For Authors:**

1. Can the authors clarify in what precise sense the proposed synthetic data are “unbiased" ?
2. The authors should discuss the privacy claims in detail, including privacy analysis under different threat models, and empirical results.

**Limitations:**

The authors do not discuss the limitations in the manuscript.

**Strengths And Weaknesses:**

**Strengths**
1. The paper studies an interesting problem in FL. The idea of improving synthetic data quality to mitigate client heterogeneity provides some insights.
2. The paper proposes to separate principal and residual components in representations. The decomposition perspective is potentially useful for synthetic data optimization.
3. The paper also includes both theoretical and empirical analysis.



**Weakness**
1.The notion of “unbiased” is not clearly defined. The paper repeatedly emphasizes “unbiased” synthetic data, but does not provide a sufficiently clear and precise explanation of what this term means in the context of the proposed method. As a result, several claims appear overstated. For example, “unbiased synthetic data and thus convergence” in the manuscript is not validated. The results support improved optimization quality, but not necessarily such strong conclusions.
2. The theoretical analysis relies on strong simplifications. The argument is built on feature-space decomposition assumptions that may not hold in complex deep representation spaces. The paper should discuss these more carefully.
3.The experiments are restricted to image classification and compare only with a small number of aggregation-free baselines. More representative state-of-the-art baselines and broader tasks are needed.
4.The privacy claim is insufficiently validated. Although the paper claims differential privacy, it does not provide corresponding privacy-focused experiments, such as privacy-utility trade-off evaluations.


**Suggestions**
There are several issues in presentation, including typos, unclear explanations, and inconsistent notations. For example, the definition and usage of T appear multiple times with potentially different roles, which may cause confusion.

---

> ### Author Rebuttal · Authors · 2026-03-31
>
> Dear reviewer, thank you for recognizing the contributions of our paper and your valuable feedback. We sincerely hope our responses address your concerns and we look forward to engaging in constructive discussion with you.
>
> 1.	Precise meaning of “unbiased”.
> We agree that the current manuscript uses the term “unbiased” too broadly. In our paper, it should be interpreted more precisely as reducing the optimization bias introduced by mean-based principal matching, rather than claiming full distributional unbiasedness in all moments. The formally established part of our theory is narrower: under Assumption 3.1, HOB yields a lower rank-1 reconstruction error than the mean for representing the principal component, which motivates using HOB instead of mean for principal matching. Based on this, we further use feature-wise variance as a lightweight proxy for residual dispersion, thereby preventing the synthetic data from becoming overly concentrated around the class center. We will revise the wording throughout the paper to make this scope explicit and avoid statements that sound stronger than what is formally proved.
> 2.	On theory simplifications and feature-space assumptions.
> We agree that Assumption 3.1 is a simplifying assumption, and that the current draft should discuss its scope more carefully. In FedUSD, the hidden representations are extracted from the penultimate-layer feature space, where representations are already highly correlated with class-discriminative information, we will clarify the details explicitly in the revision. Therefore, samples from the same class tend to share a dominant prototype direction [1], and their spectral energy is typically concentrated in the first orthogonal basis, regardless of the complexity in the feature space. Beyond the Fig. 3, we also examine this phenomenon on several mainstream datasets and networks, and observe the same trend consistently (See https://drive.google.com/file/d/1uruvgBPMALj_6gQRqZkevBHSMXzTP6Vw/view?usp=sharing). These results further support the reasonableness of Assumption 3.1.
> [1] Yang Y, Yuan H, Li X, et al. Neural Collapse Inspired Feature-Classifier Alignment for Few-Shot Class-Incremental Learning. ICLR 2023.
> 3.	Breadth of experiments and baselines.
> To strengthen the comparisons, we further take FedDM as the AFFL baseline and replace its synthetic-data optimization module with other dataset distillation methods, including M3D (AAAI 2024) and G-VBSM (CVPR 2024). In addition, we further include experiments on an audio classification task using the Speech Commands-8 dataset. The overall experimental setup is kept similar to that on CIFAR-10. For the Speech Commands-8 dataset, we first convert each audio sample into a log-Mel spectrogram and then apply SEA to the resulting time-frequency representation in the same manner as for image inputs. These new experiments further support the effectiveness of FedUSD under a broader set of comparisons.
>
> FedDM+:
> | Methods | α=0.01 | α=0.05 | α=0.1 |
> |---|---:|---:|---:|
> | FedDM+M3D | 63.18 | 65.72 | 67.09 |
> | FedDM+G-VBSM | 61.15 | 62.53 | 64.09 |
> | **FedUSD** | **67.67** | **71.05** | **74.73** |
> Speech Commands-8:
>
> | Methods | α=0.01 | α=0.05 | α=0.1 |
> |---|---:|---:|---:|
> | FedDM | 74.22 | 77.51 | 78.35 |
> | FedAF | 76.65 | 78.05 | 81.03 |
> | **FedUSD** | **82.35** | **84.18** | **85.74** |
>
> 4.	Privacy claim and privacy-focused experiments.
> To further address this concern, we further evaluate the privacy-utility trade-off of FedUSD on CIFAR-10 with ConvNet by incorporating DP-SGD under different privacy budgets when $\alpha=0.1$. As privacy strengthens (larger $\sigma_{DP}$), accuracy gradually drops, while FedUSD still maintains strong utility at moderate noise levels.
> At the same time, the appendix analyzes instance-level leakage for the DP-SGD version of FedUSD, not claiming that uploading synthetic data removes all practical privacy risks. In particular, under an honest but curious server, label-distribution leakage may still occur if the uploaded synthetic set reveals which classes are present or missing on a client. A simple mitigation is to enforce a fixed upload quota and fill missing classes with dummy synthetic data that are excluded from training. We will clarify these privacy analysis under different threat models in the revision.
> | Methods | σ_DP=1 | σ_DP=3 | σ_DP=5 |
> |---|---:|---:|---:|
> | FedDM | 63.42 | 61.51 | 58.17 |
> | FedAF | 66.92 | 65.64 | 63.21 |
> | **FedUSD** | **73.96** | **72.42** | **70.38** |
> 5.	Presentation issues and notation.
> We appreciate this comment and will carefully revise the presentation. In particular, we will correct typos, improve unclear explanations, unify notation, and explicitly disambiguate symbols such as T, which currently appears in multiple contexts.
> 6.	Limitations.
> This work mainly validates the core method under a unimodal setting. In future work, we will further investigate and verify its applicability toward cross-modal tasks.

---

> > ### Author Rebuttal · Reviewer_45bK · 2026-04-02
> >
> > Thanks to the authors for their explanations. I tend to maintain my score.

---

> > > ### Author Response · Authors · 2026-04-02
> > >
> > > Thank you for the valuable feedback and for recognizing our responses. We will include all supplemented experimental results and discussion in the revised paper. We also sincerely appreciate your recognition of our motivation and theorem in the Paper Strengths section. Should there be any further questions regarding the novelty, we would be glad to provide additional clarification.

---

### Official Review · Reviewer_iE5P · 2026-03-12

**Soundness:** 3
**Presentation:** 3
**Significance:** 2
**Originality:** 2
**Overall Recommendation:** 4
**Confidence:** 4

**Summary:**

This paper introduces an aggregation-free federated learning approach, FedUSD, to synthesize and share small class-balanced datasets from clients to the server for global model training. The core idea of FedUSD is to decompose principal and residual components of class conditional features by matching the top singular vector of the feature matrix. The paper also introduces an interesting theory where the authors argue that HOB is a better rank 1 representative than the mean.

**Compliance With Llm Reviewing Policy:**

Affirmed.

**Key Questions For Authors:**

Check Weaknesses

**Limitations:**

Yes

**Strengths And Weaknesses:**

**Strengths**


* The paper tackles an interesting problem, which is whether synthetic data based FL can reduce heterogeneity more than gradient model aggregation

* The empirical results show significant improvement. In all scenarios, FedUSD bascially outperforms all selected baselines.

* The high level motivation of this paper is easy to follow.

* The paper also introduces an interesting theory


**Weaknesses**


* The novelty of this paper is a bit limited what stands out here isn't entirely new. Though the core idea of aggregtation free federated learning remains the same as prior work and the key change is the distillation metric which is basically replacing the mean based principal.

* What looks like an argument for unbiased synthetic data goes beyond the actual proof offered. The main theory is based on assuming energy concentration $\[\sigma_1 \gg \sigma_m \quad \text{for } m \neq 1\]$ in the first singular direction and on variance residuals. This is a nice observation but it doesn't prove that the full synthetic data is unbiased at all. The independence between these matched statistics is not proved at all and using only feature wise variance basically ignores how features covary across dimensions.

* The argument of differential privacy (DP) is completely unconvincing. There is no randomized mechanism or noise calibration mentioned. so basically the standard $(\epsilon,\delta)$ is unlikely to hold given that the synthesis approach follows a fixed predictable pattern.


* The HOB is proposed by the authors using the SVD of the hidden representation matrix given as $Z_{ik}=U_{ik}\Sigma_{ik}V_{ik}^{\top}$ and then it is mentioned that the first orthogonal basis $v_{ik}$ and $\tilde{v}_{ik}^{t}$

are the HOBs in the loss $L_{\text{HOB}}$. The issue is singular vectors are unique only up to a sign, which means $v$and $-v$ describe identical dircetion, but the paper does not address any sign fixing at all before comparing with EMD. That basically means two mathematically equivalent directions could cause large loss if one returns $v$ and the other returns $-v$. I also find the use of EMD is underexplained or unclear to me. What is $M$ in Equation 14 and why is EMD preferable to sign invariant cosine or $L_2$ alignment for the direction matching?

---

> ### Author Rebuttal · Authors · 2026-03-31
>
> Dear reviewer, thank you for recognizing the contributions of our paper and your valuable feedback. We sincerely hope our responses address your concerns and we look forward to engaging in constructive discussion with you.
>
> 1.	Novelty beyond prior AFFL methods.
> Thank you for your suggestions. We agree that FedUSD shares the AFFL paradigm with prior work, and our novelty is not in changing the high-level training protocol itself. We respectfully clarify that our contribution lies precisely in proposing a new synthetic data optimization objective. In fact, our convergence analysis of AFFL versus FL suggests that, under the aggregation-free paradigm, performance improvement should primarily come from better optimization of the synthetic data, rather than from modifying the aggregation protocol itself. It indicates that the design of the synthetic-data objective is the core methodological issue in this line of work. Based on this insight, FedUSD introduces a new objective that replaces mean-based principal matching with HOB-based principal matching.
> 2.	Scope of the “unbiased synthetic data” claim.
> Thank you for your suggestions. We acknowledge the doubt of reviewers: the current manuscript does not prove that the full synthetic data are unbiased in a strict distributional sense, manuscript does not formally prove independence between the matched statistics, and feature-wise variance does not explicitly capture cross-dimensional covariance. However, our method does not rely on independence as a prerequisite. More precisely, the term “unbiased” in our paper should be understood in a narrower and relative sense: it refers to reducing the optimization bias caused by mean-based principal matching, rather than claiming full distributional unbiasedness. The part we formally prove is also narrower: under Assumption 3.1, HOB gives a lower rank-1 reconstruction error than the mean for representing the principal component. This is why we use HOB for principal matching, while variance is used only as a lightweight proxy for residual dispersion, thereby preventing the synthetic data from becoming overly concentrated around the class center. Therefore, our current theory supports a better-motivated matching objective, rather than a full characterization of the entire feature distribution. We will revise the paper to clarify this scope and discuss covariance-aware residual modeling in future work.
> 3.	Differential privacy clarification.
> Thank you for your suggestions. Appendix G does include a randomized mechanism and the corresponding noise calibration for the DP analysis, including the Gaussian mechanism, the DP-SGD version of FedUSD, parallel composition, and post-processing.
> We agree that this part is not emphasized clearly enough in the main paper. Our DP claim is intended for the DP-SGD version of FedUSD under the conditions specified in Appendix G, rather than for the deterministic synthesis procedure without added noise. We will clarify this scope in the revision and further discuss practical privacy issues.
> In addition, we further evaluate the privacy-utility trade-off of FedUSD on CIFAR-10 with ConvNet by incorporating DP-SGD under different privacy budgets when $\alpha=0.1$. The results in the table below show that FedUSD still maintains strong performance under moderate privacy budgets.
> | Methods | σ_DP=1 | σ_DP=3 | σ_DP=5 |
> |---|---:|---:|---:|
> | FedDM | 63.42 | 61.51 | 58.17 |
> | FedAF | 66.92 | 65.64 | 63.21 |
> | **FedUSD** | **73.96** | **72.42** | **70.38** |
>
> 4.	Sign ambiguity and EMD clarification.
> Thank you for your suggestions. We agree that the current manuscript does not explicitly address the sign ambiguity of singular vectors. Since $v_{ik}$ and $-v_{ik}$ represent the same direction, our implementation already uses a sign-invariant matching treatment before computing $\mathcal{L}_{HOB}$, we agree that this is not stated explicitly in the current manuscript, and we will clarify it in the revision. In Eq. (14), $\mathcal{M}$ denotes the dimension of the HOB / variance vector being matched. As for EMD, our intention is to treat HOB and variance as feature-space distributions and match their overall shape rather than only their pointwise difference. The benefit is that EMD captures global mass relocation across feature dimensions, making it more sensitive to the overall feature-distribution structure. This is necessary because deep features often exhibit coupling and redundancy across dimensions, thus, exact coordinate-wise correspondence is not always the most important factor. Instead, the overall way that energy is distributed across feature dimensions often better reflects the structure of class-conditional representations. We agree, however, that for direction matching alone, a sign-invariant cosine or sign-invariant $L_2$ metric is also a reasonable choice. We will clarify this motivation more explicitly in the revision.

---

> > ### Author Rebuttal · Reviewer_iE5P · 2026-04-02
> >
> > Thank you for the detailed rebuttal. I appreciate the authors’ clarifications, especially regarding the scope of the “unbiased” claim, the fact that the DP guarantee applies to the DP-SGD version rather than the deterministic synthesis procedure, and the explanation of the sign-invariant treatment in the HOB matching. These responses address several of my main concerns and improve my confidence in the technical framing of the paper.
> >
> > I still view the novelty as somewhat incremental relative to prior AFFL methods, since the main advance is a stronger synthetic-data objective within the same overall paradigm. However, the rebuttal makes the contribution and its scope clearer, and overall I will keep my positive evaluation and maintain my score.

---

> > > ### Author Response · Authors · 2026-04-02
> > >
> > > Thank you for the valuable feedback and for recognizing our responses. We will include all supplemented experimental results and discussion in the revised paper. We also sincerely appreciate your recognition of our motivation and theorem in the Paper Strengths section.

---

### Official Review · Reviewer_zqVR · 2026-03-18

**Soundness:** 2
**Presentation:** 2
**Significance:** 2
**Originality:** 2
**Overall Recommendation:** 3
**Confidence:** 3

**Summary:**

The problem data heterogeneity (Non-IID distributions) in Federated Learning (FL) is an important issue which has been discussed in various prior works. In the paper the authors look into ways that we can alleviate this data heterogeneity using synthetic data and they propose FedUSD, an Aggregation-Free FL (AFFL) methodology. The authors identify a key flaw in existing dataset distillation methods for FL: using the mean of hidden representations fails to separate the principal and residual components of the data, introducing bias. To resolve this, FedUSD uses the High-energy Orthogonal Base (HOB) to extract the class-level principal component and uses variance to capture the residual component. The synthetic data is then optimized on the client side by matching the HOB and variance of the real data using Wasserstein-1 Distance (EMD). Finally instead of training the model locally, now the clients need to send the synthetic data to server where one global model gets updated using all the synthetic data.

**Compliance With Llm Reviewing Policy:**

Affirmed.

**Key Questions For Authors:**

* Can you provide the empirical results for FedDM + SEA and FedAF + SEA?

* How does FedUSD perform on datasets where the energy is not heavily concentrated in the first orthogonal basis? Are there scenarios where you would need to match the top-$k$ orthogonal bases instead of just rank-1?

* Can you elaborate more on the privacy of this method, especially about the privacy metrics that achieve a good performance?

**Limitations:**

The authors do not discuss anything about the limitations.

**Strengths And Weaknesses:**

**Strength**
- Shifting from mean-matching to HOB-matching for dataset distillation in FL is a logically sound approach. This is technique is done based on an observation that energy consistently concentrates in the first orthogonal basis (Figure 3).
- The performance improvements over the provided baselines are substantial.
- The paper makes a compelling practical case for AFFL. Table 3 and Table 4 effectively demonstrate that transmitting small synthetic datasets is vastly more communication-efficient than broadcasting full model weights.
- The paper is mostly easy to follow and different steps are explained properly.

**Weakness**
- One of the main components of this paper is based on assumption 3.1. While Figure 3 makes a compelling case for this assumption but it is not very clear that the results are artifact of this specific dataset and model or can be generalized to other cases as well.

- The number of clients is set to be 10. This may mean that such techniques only work when the clients have a large local dataset. However, this is not true for low resource scenarios and might not be realistic.

- The major problem with AFFL is that it has privacy problems. By sharing the synthetic data with the server the users are essentially sharing the label distribution and there is not guarantee that the synthetic data themself might reveal some important information. I saw that in section G in appendix, the author talk about DP-SGD but based on my understanding even with DP-SGD sharing the synthetic data with the server is still problematic.

- Table 5 reveals an critical question in the experimental narrative. For CIFAR-10 ($\alpha=0.01$), the baseline is 56.30%. Adding only SEA raises this to 64.50% (an 8.2% jump). Adding only USD Loss to the baseline yields 65.41%. FedUSD (both combined) yields 67.67%. A significant portion of FedUSD's superiority over FedDM/FedAF comes from the SEA module, not just the HOB-matching. The authors failed to include a baseline of FedDM + SEA or FedAF + SEA. Without this, we cannot isolate the true delta provided by the HOB/variance matching over mean matching.

- The field of FL with synthetic data generation to tackle heterogeneity is actually very popular. However for baselines, the authors mainly consider the old aggregation methods for heterogenous setting.

---

> ### Author Rebuttal · Authors · 2026-03-31
>
> Dear reviewer, thank you for recognizing the contributions of our paper and your valuable feedback. We sincerely hope our responses address your concerns and we look forward to engaging in constructive discussion with you.
>
> 1.	Generality of Ass. 3.1.
> Thank you for your suggestions. Our evidence clarifies that Ass. 3.1 is not restricted to a specific dataset-model pair. In FedUSD, the hidden representations are extracted from the penultimate-layer feature space, where representations are already highly correlated with class-discriminative information, we will clarify the details explicitly in the revision. Therefore, samples from the same class tend to share a dominant prototype direction [1], and their spectral energy is typically concentrated in the first orthogonal basis. Beyond the example shown in Fig. 3, we also examine this phenomenon on several mainstream datasets and networks, and observe the same trend consistently (See https://drive.google.com/file/d/1uruvgBPMALj_6gQRqZkevBHSMXzTP6Vw/view?usp=sharing). When the energy is not strongly concentrated in the first orthogonal basis, a natural extension is to match the top-k orthogonal bases. We do not observe such a case in our current benchmarks, but we agree it an important extension and will discuss it in the revision.
> [1] Yang Y, Yuan H, Li X, et al. Neural Collapse Inspired Feature-Classifier Alignment for Few-Shot Class-Incremental Learning. ICLR 2023.
> 2.	Number of clients and realism in lower-resource settings.
> Thank you for your suggestions. To address this concern, we add an experiment on CIFAR-10 with ConvNet under a 100-client setting, where 10 clients are sampled in each round to perform dataset distillation, while all other settings remain the same as in the main paper. The results show that FedUSD still maintains a distinct advantage over other methods.
> We would also like to clarify that FedUSD is not restricted to scenarios where each client owns a large local dataset. As described on page 4, left column, lines 185-187 of the paper, for a class to be distilled on a client, its selected sample size only needs to be greater than IPC. In our experiments, IPC is typically small (usually below 10).
> | Methods | α=0.01 | α=0.05 | α=0.1 |
> |---|---:|---:|---:|
> | FedAvg | 36.26 | 51.84 | 55.63 |
> | FedDM | 56.84 | 59.21 | 61.07 |
> | FedAF | 59.33 | 62.48 | 64.15 |
> | **FedUSD** | **64.74** | **68.12** | **71.46** |
> 3.	Privacy concern.
> Thank you for your suggestions. We agree that the potential privacy risk of label-distribution leakage is not explicitly discussed in the current manuscript. To mitigate this issue, a simple way is to enforce a fixed upload quota for every client. In particularly, missing classes are filled with dummy synthetic data that are not used for training. We will clarify it in the revision.
> In addition, we further evaluate the privacy-utility trade-off of FedUSD on CIFAR-10 with ConvNet by incorporating DP-SGD under different privacy budgets when $\alpha=0.1$. As shown in table, FedUSD still maintains strong utility under moderate privacy budgets.
> | Methods | σ_DP=1 | σ_DP=3 | σ_DP=5 |
> |---|---:|---:|---:|
> | FedDM | 63.42 | 61.51 | 58.17 |
> | FedAF | 66.92 | 65.64 | 63.21 |
> | **FedUSD** | **73.96** | **72.42** | **70.38** |
> 4.	Ablation study.
> Thank you for your suggestions. We add the requested FedDM/ FedAF +SEA  experiments on CIFAR-10 with ConvNet under the same settings as in the main paper. These additional controls help verify that the gain of FedUSD cannot be fully explained in SEA alone. At the same time, we would like to clarify that the more direct evidence is provided by our existing +USD Loss ablation. As already discussed in the paper (page 7, left column, lines 380-384), even using only $\mathcal{L}_{USD}$ without SEA, our method already outperforms the compared methods. Therefore, the newly added results serve as complementary evidence, while the independent contribution of the proposed matching strategy is more directly reflected by the +USD Loss ablation in Table 5.
> | Methods | α=0.01 | α=0.05 | α=0.1 |
> |---|---:|---:|---:|
> | FedDM+SEA | 64.24 | 68.87 | 71.13 |
> | FedAF+SEA | 65.91 | 70.02 | 72.04 |
> | **FedUSD** | **67.67** | **71.05** | **74.73** |
> 5.	Breadth of baselines and positioning.
> Thank you for your suggestions. To strengthen the comparisons, we further take FedDM as the AFFL baseline and replace its synthetic-data optimization module with other dataset distillation methods, including M3D (AAAI 2024) and G-VBSM (CVPR 2024). These additional results further support the effectiveness of FedUSD.
> | Methods | α=0.01 | α=0.05 | α=0.1 |
> |---|---:|---:|---:|
> | FedDM+M3D | 63.18 | 65.72 | 67.09 |
> | FedDM+G-VBSM | 61.15 | 62.53 | 64.09 |
> | **FedUSD** | **67.67** | **71.05** | **74.73** |
> 6.	Limitations.
> This work mainly validates the core method under a unimodal setting. In future work, we will further investigate and verify its applicability toward cross-modal tasks.

---

> > ### Author Rebuttal · Reviewer_zqVR · 2026-04-06
> >
> > The authors have addressed my questions.

---

> > > ### Author Response · Authors · 2026-04-06
> > >
> > > Thank you for the valuable feedback and for recognizing our responses. We will include all supplemented experimental results and discussion in the revised paper. We also sincerely appreciate your recognition of our motivation in the Paper Strengths section. Should there be any further questions, we would be glad to provide additional clarification.

---

### Decision · Program_Chairs · 2026-04-30

**Decision:**

Accept (regular)

**Comment:**

This paper received borderline reviews (3, 4, 4, 4). Reviewers raised concerns on novelty, presentation, privacy, and experiments. The authors’ rebuttal helped a lot. More specifically, reviewer zqVR (score 3) indicates the concerns are fully resolved. Reviewer iE5P highlighted the remaining concern on technical novelty and incremental nature, while giving a borderline positive assessment (4). Reviewer 45bK and hoBj are both satisfied with the rebuttal (after the second round). Overall, AC agrees with the reviewers assessment: this is an incremental work with no obvious flaws. While both AC and reviewers appreciate the extensive experiments, particularly the additional results during the short rebuttal period, there are two major caveats of this paper: using CIFAR-10 and CIFAR-100 as the primary experimental datasets significantly limited the reliability of the scientific insights; as reviewers mentioned, there are many synthetic data work in federated learning (and differential privacy), see for example [how to dp-fy your data](https://arxiv.org/abs/2512.03238) Section 6, which should be discussed. Considering both pros and cons, AC would recommend acceptance if the space allows.